# Pan-cancer multi-omic model of LINE-1 activity reveals locus heterogeneity of retrotransposition efficiency

Alexander Solovyov[1,8] ✉, Julie M. Behr [2,8], David Hoyos[1], Eric Banks[2,3], Alexander W. Drong[2], Bryan Thornlow[2], Jimmy Z. Zhong[2], Enrique Garcia-Rivera[2], Wilson McKerrow[2], Chong Chu [2], Cedric Arisdakessian [2], Dennis M. Zaller[2], Junne Kamihara [4,5,6], Liyang Diao[2,9], Menachem Fromer [2,9] & Benjamin D. Greenbaum [1,7,9] ✉

Somatic mobilization of LINE-1 (L1) has been implicated in cancer etiology. We analyzed a recent TCGA data release comprised of nearly 5000 pan-cancer paired tumor-normal whole-genome sequencing (WGS) samples and ~9000 tumor RNA samples. We developed TotalReCall an improved algorithm and pipeline for detection of L1 retrotransposition (RT), finding high correlation between L1 expression and "RT burden" per sample. Furthermore, we mathematically model the dual regulatory roles of p53, where mutations in *TP53* disrupt regulation of both L1 expression and retrotransposition. We found those with Li-Fraumeni Syndrome (LFS) heritable *TP53* pathogenic and likely pathogenic variants bear similarly high L1 activity compared to matched cancers from patients without LFS, suggesting this population be considered in attempts to target L1 therapeutically. Due to improved sensitivity, we detect over 10 genes beyond *TP53* whose mutations correlate with L1, including *ATRX*, suggesting other, potentially targetable, mechanisms underlying L1 regulation in cancer remain to be discovered.

More than half of the human genome is composed of repeat sequences[1–3]. Normally, epigenetic repression and other processes silence many repeats[4] but oncogenesis disrupts these pathways[5,6]. In cancer, repeats can be re-expressed as RNA, translated in some cases into protein, and may be actively involved in genome instability and cancer immunogenicity[7–12]. The Long INterspersed Element-1 (LINE-1 or L1) transposable element (TE) is an especially interesting class of repeats, possessing the ability to reinsert itself via retrotransposition at new loci in the human genome and parasitizing the genome in the process[13]. An intact L1 element is ~6 kb in length, but most of the

>500,000 copies that comprise ~20% of the human reference genome are truncated upstream of their 3' ends[1]. There are just over 100 L1s capable of coding for the full-length ORF1p and ORF2p proteins, the latter containing the reverse transcriptase and endonuclease[1] needed for L1 activity. Of note, L1 RNA not only encodes the protein machinery for retrotransposition but also acts as the template for new genomic copies of L1.

Understanding the functional significance of L1 in cancer requires quantifying its activity across its life cycle, including retrotransposition[11,14–19] and expression. Further, intact L1 elements

[1]Halvorsen Center for Computational Oncology, Department of Epidemiology and Biostatistics, Memorial Sloan Kettering Cancer Center, New York, NY, USA. [2]ROME Therapeutics, Inc., Boston, MA, USA. [3]Acorn Biosciences, Cambridge, MA, USA. [4]Division of Hematology/Oncology, Boston Children's Hospital, Boston, MA, USA. [5]Department of Pediatric Oncology, Dana-Farber Cancer Institute, Harvard Medical School, Boston, MA, USA. [6]Division of Population Sciences, Dana-Farber Cancer Institute, Harvard Medical School, Boston, MA, USA. [7]Physiology, Biophysics & Systems Biology, Weill Cornell Medical College, New York, NY, USA. [8]These authors contributed equally: Alexander Solovyov, Julie M. Behr [9]These authors jointly supervised this work: Liyang Diao, Menachem Fromer, Benjamin D. Greenbaum. ✉e-mail: solovyova@mskcc.org; greenbab@mskcc.org

that reside at different genomic loci can plausibly exhibit heterogeneous activity[14,20,21], akin to genes that have arisen from duplication events, whose functions may diverge as a result of mutational processes and selection that act mostly independently on the copies[22,23]. The literature already contains ample evidence that L1 loci vary in some ways, including in their levels of retrotransposition ("hot" loci being those that are more RT-active[11,15,17,18,24–27]), RNA expression levels[14,21], and even allelic variability at the same genomic locus, resulting in variable activity[20]. However, it is generally parsimoniously assumed that the in vivo retrotransposition rates of intact L1 elements are proportional to their RNA expression levels (arising, e.g., from local differences in genomic regulation), rather than variable properties of the L1 RNAs and/or proteins themselves.

Though clearly active in cancer, quantifying L1 activity from short-read sequencing data poses technical challenges due to the multitude of genomic copies of highly similar L1 sequences, making it difficult to disambiguate the source of L1-containing sequencing reads. In this work, we develop and benchmark a computational pipeline for detecting somatic L1 retrotransposition events that includes the "TotalReCall" algorithm (Fig. 1), which we pair with an additional well-established caller, xTea[19], to call consensus retrotranspositions in cancer. Employing this dual pipeline, we derive high confidence retrotransposition prevalence ("RT burden"). To quantify L1 RNA expression alongside RT burden, we used the L1EM mechanistic model of L1-driven transcription ("active expression") at intact L1 copies in the genome[28].

Leveraging the recent large-scale release of The Cancer Genome Atlas (TCGA) whole-genome sequencing and RNA-sequencing data from tumor and normal samples, we quantify somatic L1 RT burden in a pan-cancer cohort of 4669 paired tumor-normal samples across 31 tumor types. We find a high correlation between L1 RNA expression in tumor samples and the burden of L1 retrotranspositions that have accumulated during the historical evolution of that tumor. Additionally, we provide evidence, across a large set of loci, that the sequence products of individual loci behave heterogeneously, which both supports and adds to previous in vitro experimental reports[24].

Given their potentially pathogenic nature, it is unsurprising that there are multiple levels of regulation of L1 activity, only some of which are beginning to be understood. In our integrated omics analysis, we elucidate the intricacies of these regulatory mechanisms. For example, while mutations in *TP53* are generally associated with L1 retrotransposition activity[11,18,29–32], recent work has also shown that p53 may directly regulate L1 at the RNA level via control of its transcription[33]. Using a statistical model relying on their joint measurements, we find that even with potential regulation of L1 RNA transcription by p53, there remains a large effect of p53 on L1 RT burden that is independent of the level of L1 RNA expression. Of clinical interest, we use this dataset to assess L1 activity in tumors from individuals with Li-Fraumeni Syndrome (LFS, bearing germline *TP53* pathogenic/likely pathogenic variants), and we find L1 activity in LFS cancer patients to be comparable to that in non-LFS tumors. Finally, expanding beyond *TP53*, we identify additional genes whose mutations are associated with L1 activity, thereby assigning these genes to biological pathways that regulate L1 in cancers.

## Results

### Many cancers are significantly enriched for L1 retrotransposition

Running TotalReCall (Fig. 1) paired with xTea (see Supplementary Methods for details on algorithms and benchmarking), we identified 64,292 somatic L1 retrotranspositions in the dataset of 4669 tumor and matched normal whole genome sequencing (WGS) paired samples. Approximately 40% of these retrotranspositions ($N = 28,210$) contain inversions of the inserted L1 sequence (Fig. 2a), consistent with the twin-priming mechanism of reintegration[34] (Fig. 1d, f). This

inversion rate was supported by both TotalReCall and xTea, and the ability of both algorithms to capture true inversions was validated using long reads from the Genome in a Bottle dataset[35] (see "Validating retrotransposition detection by TotalReCall, xTea, and TraFiC-mem using long reads", Supplementary Methods). The rate of inversion-containing insertions across cancer indications is similar, with the percentage being lowest in prostate cancer and highest in uterine tumors (Fig. 2b). In all cases, the rate of inversion-containing somatic LINE-1s is consistently higher than what has been observed in the germline[18]. Of the 36,082 canonical insertions (non-inversion containing, consistent with single-strand priming, Fig. 1c, e), approximately 6% ($N = 2143$) contain the full-length L1 sequence, while the rest are truncated to some degree (Fig. 2c).

The purpose of the work performed here was to detect tumor-specific somatically acquired, i.e., non-germline, L1 retrotranspositions; hence we did not attempt to comprehensively identify germline retrotranspositions present in a sample but not in the reference genome. Nevertheless, as a byproduct of looking for tumor-specific calls, we did collate and evaluate a set of "pseudo" germline calls (see "Non-reference L1 insertions present in both case and control samples", Supplementary Methods), which we define as those calls present in tumor samples but not categorized as somatic due to the existence of support for the retrotransposition in the corresponding normal sample. The overall distribution of the number of such "pseudo" germline retrotranspositions, along with their lengths and allele frequencies, are similar to what has previously been reported for L1[36] (Supplementary Fig. 1a–c).

The breakdown of RT burden per sample across tumor types is consistent with previous studies, though with increased sensitivity to individual calls compared to previous efforts[11,16] (Supplementary Fig. 1d–h). We considered comparisons between each cancer type with at least 5 samples and a median of at least 1 RT per sample against a background of all cancer types with median 0 RTs per sample. All such comparisons were significant, confirming the enrichment of L1 RT burden in esophageal, colon, lung squamous cell, and head and neck cancers, previously observed in the PCAWG dataset. In addition, we find significant enrichment of RT burden in in stomach, bladder, ovarian, prostate, and uterine (both carcinosarcoma and corpus endometrial carcinoma) cancers ($p < 10^{-10}$ for two-sided Mann–Whitney $U$ tests, Fig. 2d).

Approximately 15% (9912) of the insertions include transductions of sequences beyond the 3' end of the genomic source L1. We could confidently attribute 4870 of these to their genomic source coordinates. We refer to these as transduction-bearing RTs ("TRTs"; distinguished from "TDs" used by PCAWG[11] and others, which include orphan transductions without accompanying L1 sequence). These TRTs represent 726 unique genomic sources, of which a vast majority are assumed to be polymorphic non-reference L1s ($N = 610$, 84.02%), when no L1 element resides in the genomic vicinity. We annotate each of these genomic source loci in Supplementary Data 1.

Of the 198 active genomic loci catalogued by Ebert et al.[11,15,24–27], we found retrotranspositions arising from 102, including 20 that are not sequence-resolved, 10 that had previously only shown documented activity in vitro, and 13 that had not previously been documented in vivo in cancer. One-hundred and thirty-seven loci are sequence-resolved; and, of those, 68 are present in the hg38 reference genome (34 believed to be "fixed present" throughout the population). Eighty-seven loci are included as active elements in L1EM and could therefore be compared against RNA measurements. This includes 19 loci annotated as L1PA2, and 41 loci seemingly not intact in one or both ORF domains in the reference genome sequence.

### High L1 RNA corresponds to high L1 RT burden

Across TCGA, high-quality tumor RNA-seq data is available from 8998 distinct individuals across 32 tumor types, with tumor-adjacent normal

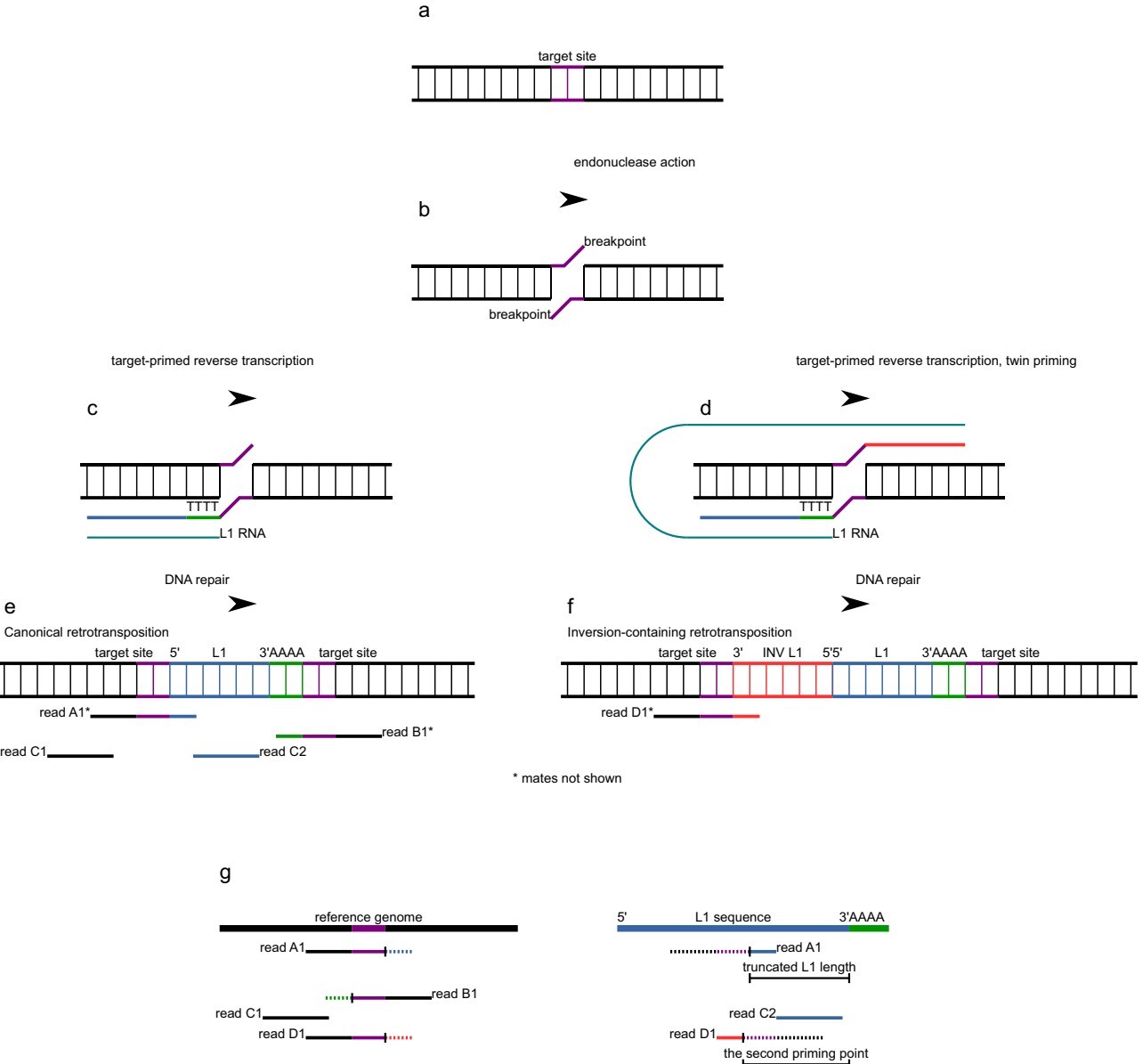

**Fig. 1 | Insertion of the LINE-1 (L1) retrotransposon and its detection by TotalReCall. a** Haploid copy of the genome before retrotransposition. **b** Endonuclease break in each strand of DNA. **c** L1 RNA directly reverse transcribed into the genome resulting in the synthesis of the single-stranded cDNA starting from the 3' end of the L1 transcript and extending a variable length towards its 5' end. Teal, L1 RNA. Green, reverse transcribed poly(T) cDNA. Blue, reverse transcribed L1 cDNA. **d** Twin priming results in the simultaneous reverse transcription of different parts of the L1 transcript into the two strands of the genome. Red, reverse transcribed L1 cDNA on the opposite gDNA strand. **e, f** Genome after synthesis of the second strand of DNA and repair. Components of the L1 sequence are annotated with respect to the "top" strand of the genome. Purple, target site, which is duplicated following repair. Red, newly inserted L1 sequence that was synthesized on the top strand and is therefore reverse complemented with respect to L1 RNA. Blue, newly inserted L1 sequence that was synthesized on the bottom strand. Green, newly inserted poly(A). Paired-end reads originating from the modified genomes are shown. **e** The process shown in **c** results in a (possibly 5' truncated) "canonical" retrotransposition. **f** The process shown in **d** results in an "inversion-containing" retrotransposition. The resulting genomic sequence has two L1 fragments in opposite orientations. **g** Mapping of reads A–D to the unmodified (reference) genome lacking the transposon insertion (left) and the transposon sequence (right). Left, tails of reads A1, B1, and D1 that come from the novel transposon are clipped (shown as dashed lines). Right, read C2 and the clipped tails of reads A1 and D1 align to the transposon sequence. The clipped tail of read B1 contains only poly(T). In the absence of inversion, the alignment between the clipped sequence and the transposon sequence reflects the length of the newly inserted transposon. When inversion occurs, such an alignment will only reflect the position where the second priming occurred.

data from 719 individuals across 32 tumor types (see "Methods"). We quantified locus-level L1 RNA "active" expression (transcription driven by the L1 promoter) from these 9717 samples using L1EM[28] and aggregated expression levels (transcripts per million, TPM) across loci to calculate the total relative abundance of active L1 RNA present in each sample (see "Quantification of LINE-1 expression at the locus level" in Supplementary Methods and Supplementary Fig. 2 for

simulations demonstrating the accuracy of locus-level quantification by L1EM).

Because we and others have identified retrotranspositions from source elements annotated as L1PA2 and/or without intact ORF domains in the reference genome, we chose to aggregate RNA expression from all 1483 quantified L1HS and L1PA2 elements in L1EM, intending this as a superset of all potentially functional loci. This

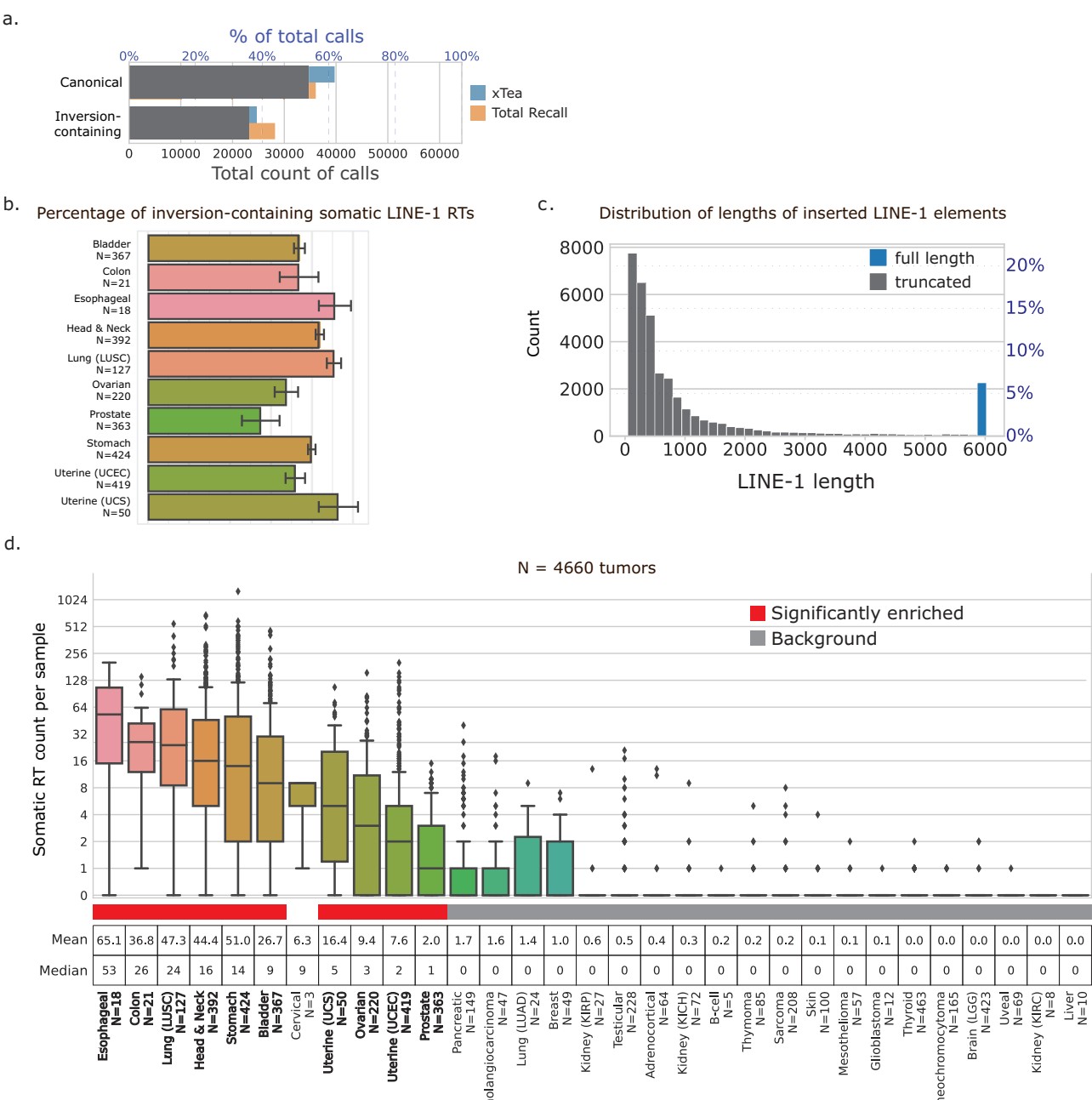

**Fig. 2 | Overview of somatic RT calls in TCGA. a** Count of calls in call set across all tumor types, annotated as either canonical or inversion-containing, as an absolute count (bottom axis) and percent of all calls (top axis). Gray bars indicate calls for which both callers share the same annotation. Blue and orange bars are additional calls annotated in each group by only xTea or only TotalReCall, respectively. **b** Percentage of insertion events containing inversions by indication for indications with ≥500 insertions total. Error bars and the bar height represent the 99% confidence interval and the estimate of the mean using the Clopper–Pearson method. **c** Estimated length of the inserted L1 within the canonical somatic RT calls. Length estimates taken from TotalReCall. Gray, truncated insertions. Blue, full-length insertions. Left axis, absolute count of calls. Right axis, percent of total canonical somatic RT calls within each bin. Length does not include transduction region, if one is present. **d** Somatic RT count per sample grouped by tumor type. Center line indicates median. Box indicates interquartile range. Points more than 1.5× IQR away from the box are shown as individual outliers. Tumor types are sorted in descending order by median somatic RT burden. Mean and median RT count per sample by tumor type listed in boxes above the tumor type name. Red bars indicate tumor types that are significantly enriched in RT (see Ns for each significantly enriched tumor type in figure), compared to a background of all samples from tumor types indicated by gray bars ($N = 2265$). All significant tumor types had $p < 10^{-10}$ from a two-sided Mann–Whitney $U$ test (effect sizes > 0.57). Cervical cancer samples also had a significant $p$-value ($p = 4 \times 10^{-8}$) from the two-sided Mann–Whitney $U$ test (effect size = 0.94) but are not considered significant here due to the small sample size ($N = 3$ tumor-normal pairs).

aggregated expression is highest in esophageal carcinoma, followed by lung squamous cell carcinoma and stomach adenocarcinoma (Fig. 3a). Lung squamous cell carcinoma has the greatest median gain of L1 expression in tumor samples with respect to adjacent-normal ($p < 1 \times 10^{-10}$, one-sided Mann–Whitney $U$ test). On the other hand,

prostate tissue has the highest levels of L1 RNA in normal samples, with at most a marginal gain in the corresponding prostate adenocarcinoma tumor samples ($p = 0.05$, one-sided Mann–Whitney $U$ test).

Expression analysis of the subset of 121 elements for which there is evidence of retrotransposition from either our analysis (87 elements)

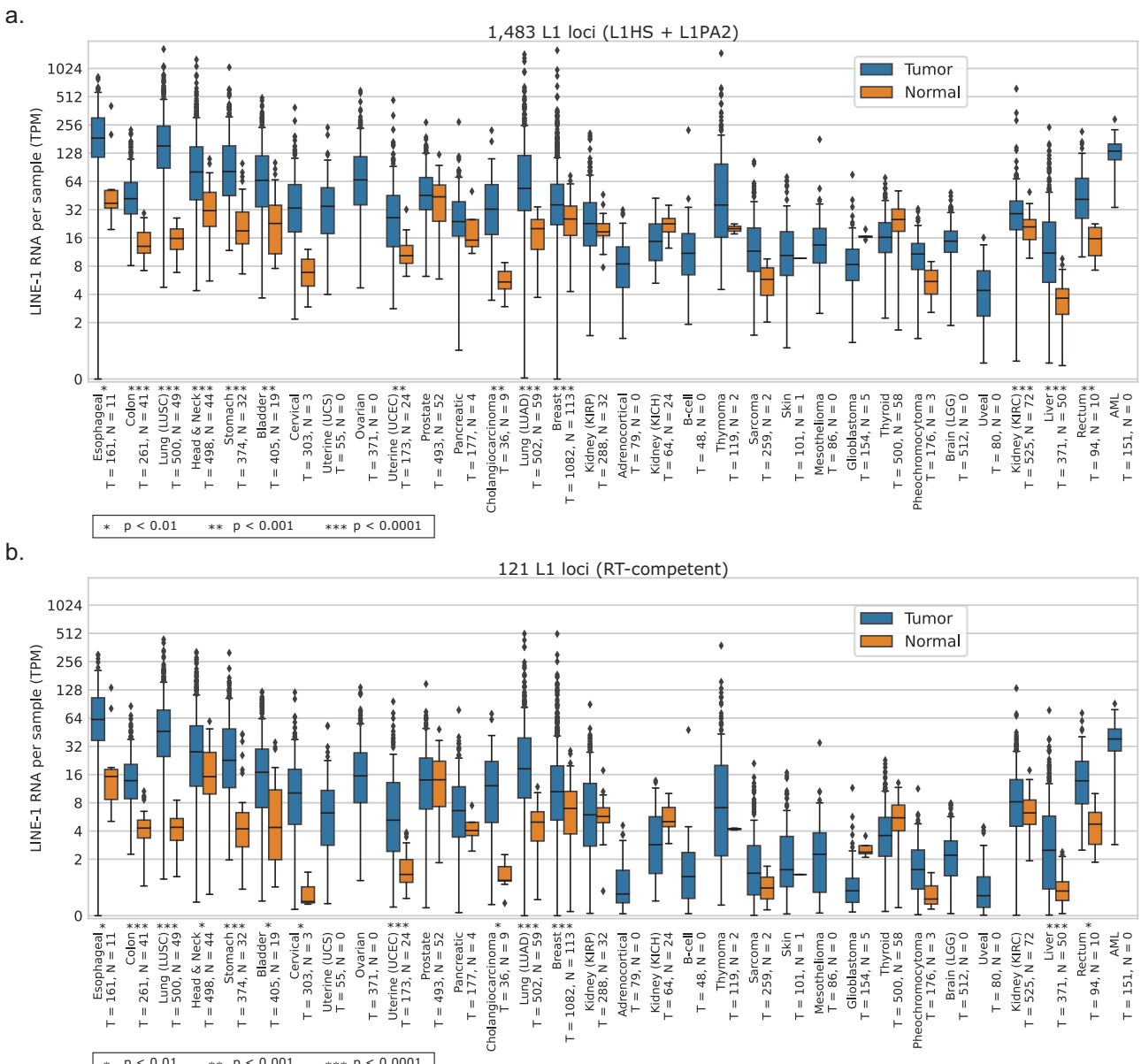

**Fig. 3 | L1 RNA throughout TCGA. a** Estimated expression of L1 RNA in each sample by tumor type, quantified by L1EM. L1 RNA expression per sample is aggregated across all active L1HS and L1PA2 loci, $N = 1483$. Asterisks (*) indicate significance level of Bonferroni adjusted one-sided Mann–Whitney $U$ test comparing tumor to normal expression (*$p < 0.01$, **$p < 0.001$, ***$p < 0.0001$). Bonferroni-adjusted $p$-values and effect sizes for the asterisked indications, from left to right in the figure, are as follows. Esophageal: $1.31 \times 10^{-3}$, 0.69; Colon: $2.88 \times 10^{-19}$, 0.90; LUSC: $1.14 \times 10^{-27}$, 0.96; Head & Neck: $5.57 \times 10^{-9}$, 0.6; Stomach: $1.57 \times 10^{-12}$, 0.79; Bladder: $2.05 \times 10^{-4}$, 0.58; UCEC: $9.06 \times 10^{-5}$, 0.56; Cholangiocarcinoma: $1.80 \times 10^{-4}$, 0.94; LUAD: $2.30 \times 10^{-20}$, 0.76; Breast: $5.57 \times 10^{-9}$, 0.33; KIRC: $1.8 \times 10^{-7}$, 0.41; Liver: $9.93 \times 10^{-14}$, 0.67; Rectum: $4.2 \times 10^{-4}$, 0.79. **b** Estimated expression of RNA from 121 L1 loci in each sample grouped by tumor type, quantified by L1EM. L1 RNA expression per sample is aggregated across 121 loci with evidence of in vitro activity or transductions (see Methods section, Identifying subsets of L1 elements). Bonferroni-adjusted $p$-values and effect sizes for the asterisked indications, from left to right in the figure, are as follows. Esophageal: $1.38 \times 10^{-3}$, 0.69; Colon: $2.47 \times 10^{-18}$, 0.87; LUSC: $1.26 \times 10^{-26}$, 0.94; Head & Neck: $5.44 \times 10^{-3}$, 0.32; Stomach: $7.4 \times 10^{-11}$, 0.73; Bladder: $9.73 \times 10^{-3}$, 0.4; Cervical: $3.49 \times 10^{-3}$, 0.94;UCEC: $2.44 \times 10^{-6}$, 0.65; Cholangiocarcinoma: $1.42 \times 10^{-3}$, 0.83; LUAD: $1.12 \times 10^{-20}$, 0.76; Breast: $4.7 \times 10^{-7}$, 0.31; Liver: $4.33 \times 10^{-9}$, 0.54; Rectum: $1.6 \times 10^{-3}$, 0.73. **a, b** Total $N = 9717$ samples; 8998 tumor samples and 719 normal samples. Tumor types are sorted as in Fig. 2, with the addition of Rectal adenocarcinoma and AML. Blue, tumor samples. Orange. normal samples. Center line indicates median. Box indicates interquartile range. Points more than 1.5× IQR away from the IQR box are shown as individual outliers.

or previously published studies (74 total, including 34 additional[26]) yielded similar results (Fig. 3b; see Supplementary Data 2 for a list of these loci). In fact, the aggregate L1 RNA expression for the superset of 1483 loci and the expression for the subset of 121 elements were nearly proportional ($R = 0.96$, $p < 10^{-10}$, Pearson correlation between all loci and active loci expression calculated from Supplementary Data 3). Because L1EM provides separate quantification for "only" (transcripts that terminate at the polyadenylation site of the L1 element itself) and

"run-on" (transcripts that extend beyond the internal polyadenylation site and include genomic sequence downstream of the 3' end of the element; also referred to as "3' extended"), we also specifically considered the expression of run-on transcripts for both sets of loci, again with similar results (Supplementary Fig. 3).

To relate L1 retrotransposition to L1 expression, we aggregated signals within tumor types and compared DNA and RNA summary statistics. The 11 tumor types with a per-sample median RT burden

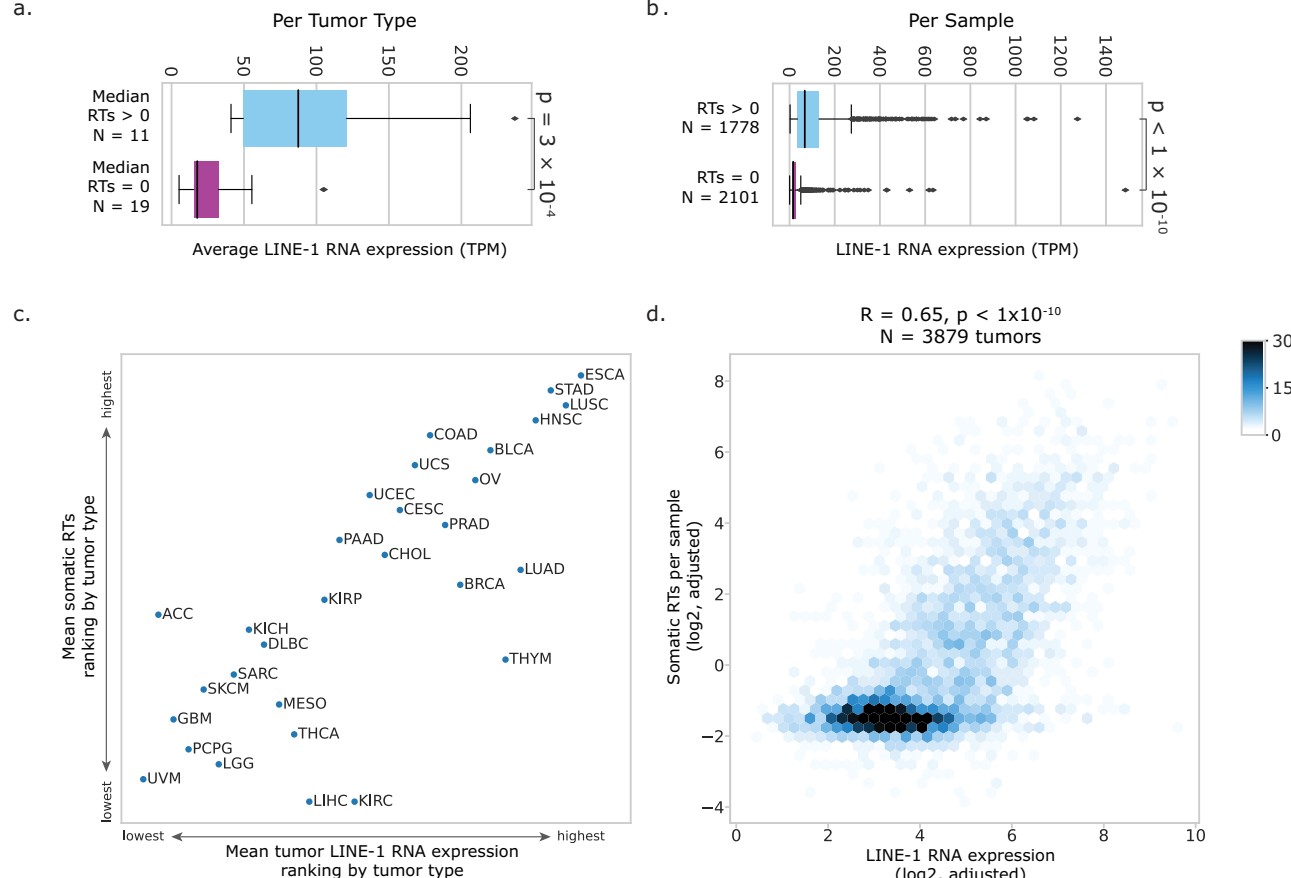

**Fig. 4 | L1 RNA throughout TCGA corresponds with RT burden. a** Average expression of L1 RNA per tumor type grouped by median RT burden per tumor type. Blue, tumor types with median RT burden > 0 per sample. Purple, tumor types with median RT burden = 0 per sample. **b** Expression of L1 RNA grouped by RT burden per sample. Blue, samples with RT burden > 0. Purple, samples with RT burden = 0. **a**, **b** Center line indicates median. Box indicates interquartile range. Points more than 1.5× IQR away from the IQR box are shown as individual outliers. *P*-value calculated from two-sided Mann–Whitney *U* test (4a: $p = 3.0 \times 10^{-4}$, effect size = 0.81, N1 = 11, N2 = 19; 4b: $p < 10^{-10}$, effect size = 0.71, N1 = 1778, N2 = 2101). **c** Comparing the relative rankings of tumor types based on mean L1 RNA expression (*x*-axis) and mean somatic L1 RT burden (*y*-axis). **d** Correlation between QC-adjusted L1 RT burden and QC-adjusted L1 RNA per tumor sample. *N* = 3879 tumor samples with both WGS and RNA-seq. *R* = 0.65, $p < 10^{-10}$, Pearson correlation (using the exact distribution, as calculated by the scipy.stats.pearsonr function in python).

above 0 have significantly higher L1 RNA expression than the 19 tumor types with a median of 0 retrotranspositions ($p < 10^{-10}$, two-sided Mann–Whitney *U* test, Fig. 4a). In total, 3879 tumor samples have both WGS and RNA-seq data, allowing us to compare RT and RNA burdens at the sample level. The tumors with a non-zero RT burden had significantly higher L1 RNA expression as well ($p < 10^{-10}$, two-sided Mann Whitney *U* test, Fig. 4b). For the 30 tumor types with at least five tumor samples in both the WGS and RNA-seq datasets, we ranked each tumor type based on average L1 RNA expression and RT burden; we found that tumor types that ranked higher based on RT burden tended to also be ranked highly based on L1 RNA (Fig. 4c). We adjusted our estimates for L1 RNA and RT burden per sample based on sequencing quality metrics to minimize technical biases that may make distinguishing biological relationships more difficult (see "Methods" for details). We found a striking correlation between L1 RT burden in a sample and its L1 RNA expression ($R = 0.65$, $p < 10^{-10}$, Pearson correlation, Fig. 4d).

**Locus-level analysis of L1**
Although only a subset of retrotranspositions can be attributed to specific progenitor elements, the size of our dataset allowed us to interrogate the relationship between their RT burden and RNA expression at the locus level. Across tumor types, most L1 elements have minimal RNA expression, and an average TRT burden of 0 (Fig. 5a-b). Among the elements with non-zero expression, there is diversity across tumor types. We clustered loci based on similar RNA and TRT patterns across tumor types (see "Methods" and Supplementary Data 3). Overall, most elements follow a similar pattern of highest-to-lowest expression as the total L1 RNA, with the highest expression in esophageal and lung squamous cell cancers (Fig. 5c), yet there are tissue-specific differences. The two highest expressed elements overall, 22q12.1 and 20p11.21-1, are highly expressed in cervical tumors, where few other loci are expressed. The 3p22.1-1 locus has a unique pattern of tumor types in which it is expressed compared to the other locus clusters, and the 3q22.1-1 cluster of 3 loci is specifically highly expressed in prostate tumors.

The landscape of locus-TRT burden across tumor types for the same clusters of loci is remarkably different (Fig. 5d). The mostly highly expressed locus, 22q12.1, has comparable expression in esophageal and cervical cancers but creates far more TRTs in cervical cancers, despite esophageal cancer having higher RT burdens overall. It also creates notable TRTs in uterine cancers (both carcinosarcoma and corpus endometrial carcinoma), despite having relatively lower expression in those tumors than many others. As another example, uterine carcinosarcomas have higher TRT burdens from Xp22.2-2 and 12p13.32-1 relative to how highly the corresponding RNAs are expressed.

Further considering such locus-level relationships between RT and RNA, we noticed that Xp22.2-2 generally creates more TRTs than would be expected given its low expression level: 22q12.1 is more

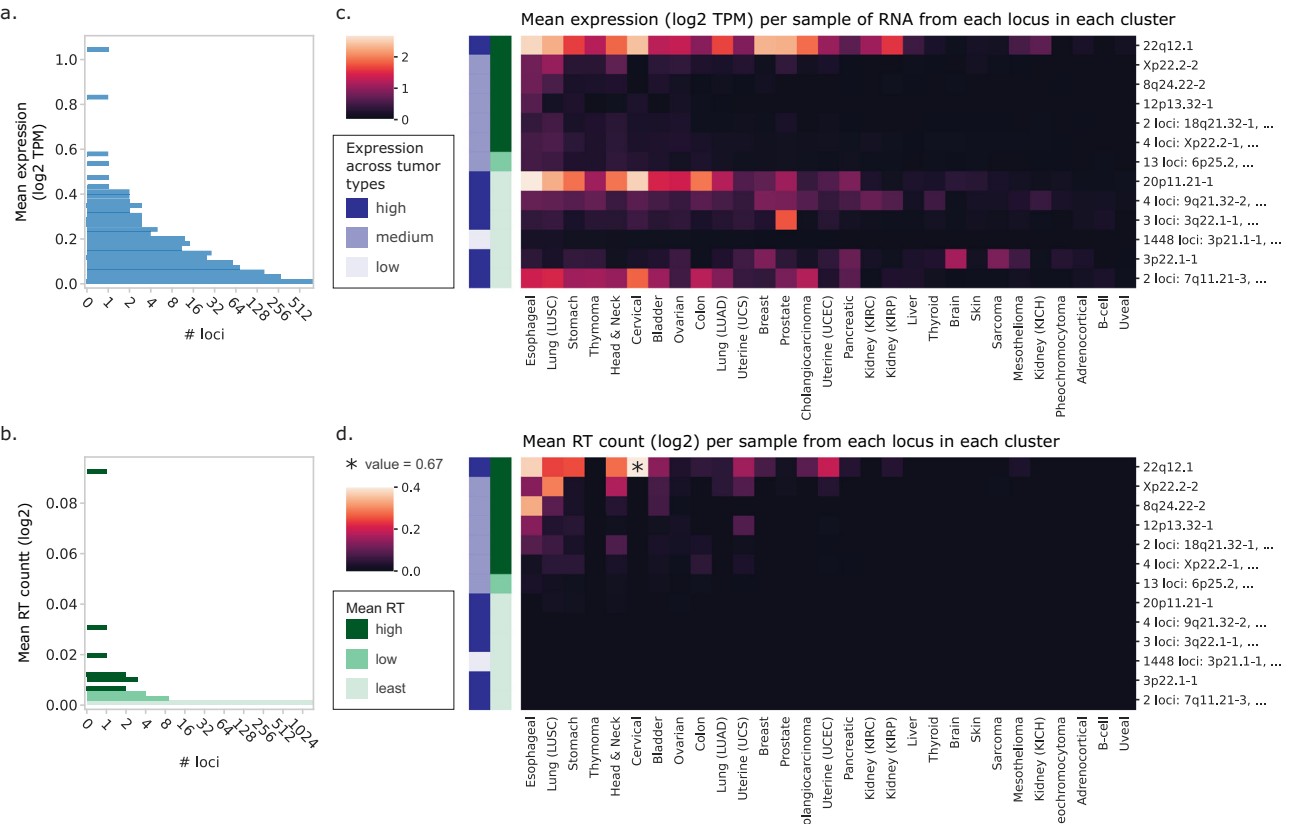

**Fig. 5 | Landscape of L1 locus activity across tumor types.** Histogram of **a** per-sample mean L1 RNA expression and **b** L1 RT, based on identified transductions, for all 1483 L1HS and L1PA2 loci. Mean expression and mean RT count per sample are weighted by the inverse of samples with the same tumor type, so each type contributes equally to the locus mean. **c** Heatmap of mean L1 RNA expression (log2 TPM) of each locus within a given cluster (rows) across tumor types (columns). **d** Heatmap of mean log2 L1 RT count (based on identified transductions) of each locus within a given cluster (rows) across tumor types (columns). The 22q12.1 locus in cervical cancer, starred, had exceptionally high RT count. To enable visualization of the variation across the heatmap, the colors were scaled to a maximum value of 0.4 and this square was marked with an asterisk to indicate its outlier value. **c, d** All 1483 L1HS and L1PA2 loci have been clustered based on similar expression and RT count profiles across tumor types, resulting in 13 clusters. Clusters are named based

on the locus in each cluster with the highest mean RNA expression and sorted from highest (top) to lowest (bottom) mean RT value. To generate each heatmap value, a mean for each locus within each tumor type is first calculated, and then the mean of means for all loci within a cluster is determined. Rows are sorted left to right by highest to lowest total L1 RNA expression (summed across all 1483 loci) per sample. To the left of each heatmap, the row colors annotate each cluster categorically based on RNA and RT. The left column (dark, medium, and light blue) indicates the distribution of RNA expression of each cluster across tumor types, with mean expression ≥0.1 TPM in >15 tumor types for "High", in 5–15 tumor types for "Medium" and in fewer than 5 tumor types for "Low". The mean RT count of all loci within the cluster was categorized into "high" (dark green), "low" (medium green), and "least" (light green) based on the histogram in **b**.

highly expressed by nearly an order of magnitude yet creates only a few more TRTs. To determine whether this could be due to detectability issues, we compared these two loci among only the subset of samples with TRTs from at least one of these two loci. Considering both total locus expression and specifically run-on expression, 22q12.1 generates proportionately much higher RNA (Supplementary Fig. 4a). Within the samples that only have TRTs from Xp22.2-2, 22q12.1 has higher expression, both in total and in run-on RNA (Supplementary Fig. 4b). Some tumors may have alleles of either locus that are not RT-competent and thereby skew the apparent ratio of RNA converting to RT. We therefore considered individual tumors with TRTs from both loci, which must have at least one functional allele of both (Supplementary Fig. 4c). The ratio of TRTs per RNA is still distinctly higher for Xp22.2-2, leading us to conclude that there are also functional differences in the activity of different RT-competent elements, referred to as the RT "efficiency" of a specific L1 element.

With the exception of Xp22.2-2, the greatest disconnect between RNA and RT competency, based on observed TRTs in this study, is in the lower few locus clusters, which have a variable pattern of RNA expression (Fig. 5c) but almost no TRTs in any tumor types (Fig. 5d), with the exception of the large cluster (N = 1448), which has both low

expression and RT activity. Many of these loci, including the latter, may not be RT-competent population-wide, and therefore could never retrotranspose regardless of RNA expression level. However, even among the 121 presumed active loci, the patterns of RNA expression and TRT burden across locus clusters and tumor types clearly are different (Supplementary Fig. 5). In theory, these differences can be reflective of tumors with active RT-disrupting mechanisms, loci that are polymorphically intact (i.e., functional in only some of the samples in which they are expressed), and variability in the fraction of run-on RNA (which would impact how many RTs resulting from each locus will be transduction-bearing and thus detectable via short-read WGS); we address some of these issues in subsequent analyses.

### Quantifying the efficiency differences between L1 loci

To determine whether there are quantifiable differences in locus efficiency, we fitted a regression model of locus TRT against locus RNA, using tumor type and p53 mutation as covariates, for 48 loci for which TRTs have been detected in at least 2 tumors (see "Methods"). In this model, the RNA coefficient is the estimated "efficiency" value, with higher values indicating greater efficiency—i.e., that even with low levels of RNA expression, a high number of RT events are observed.

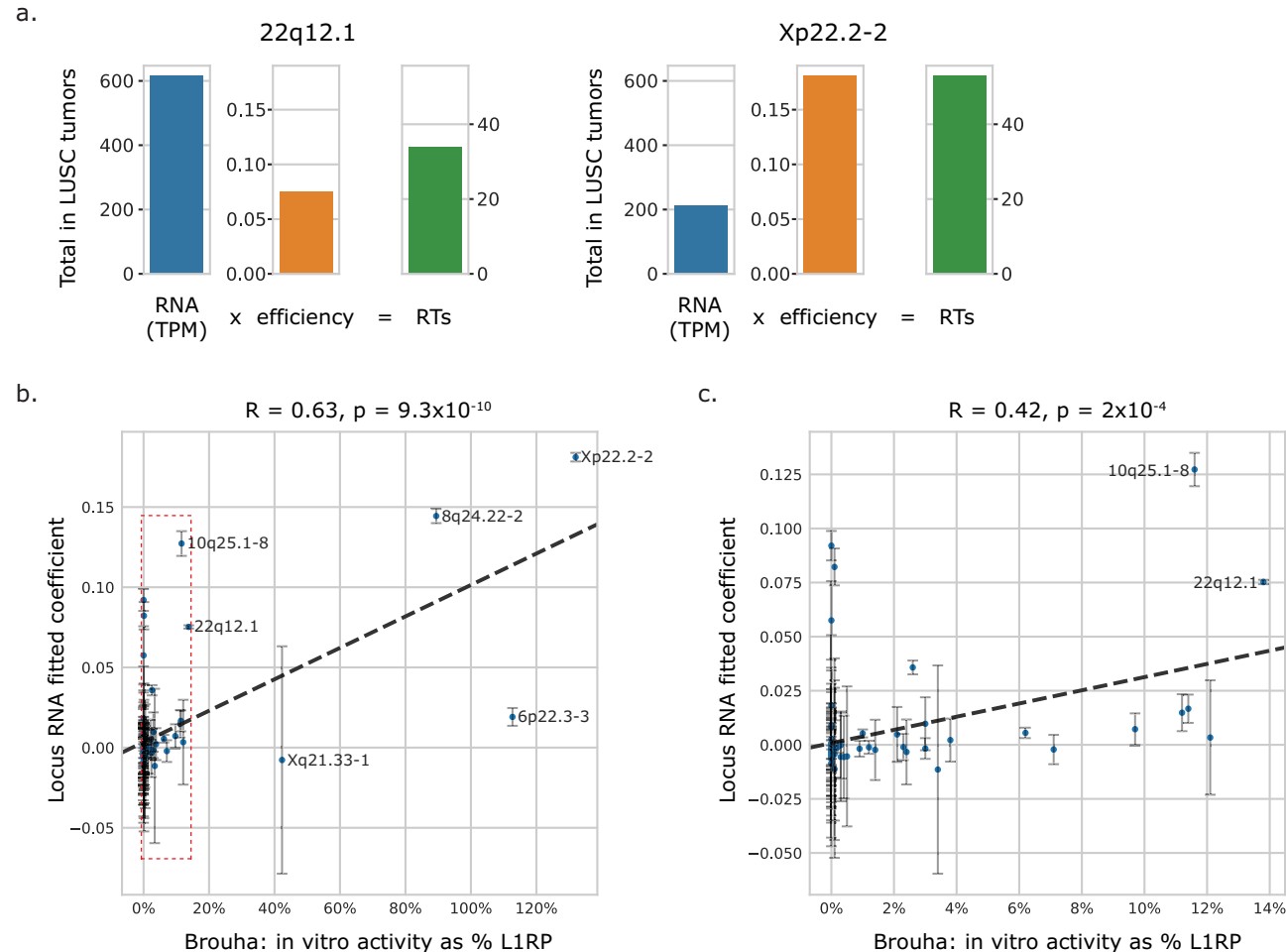

**Fig. 6 | Fitting regression models to represent locus efficiency. a** A comparison of two loci within lung squamous cell tumors (*N* = 126): sum of locus RNA expression within these samples (blue bar) scaled to the fitted efficiency value (orange bar), sum of locus TRT counts within these samples (green bar) (see "Methods" for calculation of efficiency). **b** Linear regression of coefficients assigned to individual loci (*y*-axis) vs. in vitro activity as measured by Brouha et al.[24] (*x*-axis), reported as a relative percentage of measured L1RP activity. *N* = 76 loci (subset of 156 that were measured in vitro[24]), *R* = 0.63, *p* = 9.3 × 10⁻¹⁰, Pearson correlation (using the exact distribution, as calculated by the scipy.stats.pearsonr function in python). Error bars represent 95% confidence intervals around assigned coefficient. Red box indicates region shown in (**c**). **c** Linear regression, removing 4 high-activity loci ("hot" L1s from Brouha et al.[24]). *N* = 72 loci, *R* = 0.42, *p* = 2 × 10⁻⁴, Pearson correlation (exact distribution). Similarly, error bars represent 95% confidence intervals around assigned coefficient.

The tumor type is included to account for tissue-specific differences in locus expression and general RT regulation. p53 is a known regulator of L1 RT and is included to further de-bias any apparent efficiency differences attributable to differential regulation.

To start with a concrete example in lung squamous cell tumors, we clearly see the difference noted above in RT efficiency between locus 22q12.1 and locus Xp22.2-2 (Fig. 6a). Comparing the pan-cancer efficiency estimates for every locus against a background distribution of efficiencies generated by resampling (see "Methods"), we found significantly higher variance for the actual values compared to the background for many loci (Supplementary Fig. 6a), demonstrating true heterogeneity among the RT efficiencies for the L1 loci. Comparing the efficiencies for the 121 RT-competent loci in this model to efficiency estimates to previous in vitro measurements of activity[24], we found a significant correlation (*R* = 0.63, *p* < 10⁻⁹, Pearson correlation, *N* = 76 loci, Fig. 6b), even when removing the most active loci (*R* = 0.42, *p* = 2 × 10⁻⁴, Pearson correlation, *N* = 72 loci, Fig. 6c). As expected, Xp22.2-2 has a higher estimated efficiency than 22q12.1, with the latter still moderately high (Supplementary Fig. 6a).

Although TRTs from reference loci allow us to be locus-specific, they represent only 2.5% of the total RT call set, and the efficiency estimates above do not account for activity resulting in non-

transduction bearing RTs. We therefore repeated the linear regression analysis using total RTs per sample as the response variable to locus-level RNA, thus enabling the association of all loci with RT activity, not only the ones which result in TRTs. Here we found a striking 89 and 16 elements, respectively, with significantly high and moderately high fitted coefficients (Supplementary Fig. 7). We opt to refer to these estimates as the fitted coefficients rather than efficiencies due to response variable being total RTs as opposed to locus-specific TRTs. Interestingly, some of the elements with significantly high locus-specific efficiencies, including 1p31.1-12 and Xp22.2-2, have moderately and significantly low coefficients here.

To limit a situation whereby variability in baseline expression at individual loci inflates the significance of fitted coefficients in the previous models, we next fit a model of total RT burden against the aggregate expression of each of the 13 previously generated clusters of elements (Supplementary Fig. 8), reasoning that the expression estimates for groups of similarly expressed loci would be more statistically stable. The fit of this model (*R* = 0.73, *p* < 10⁻¹⁰, Pearson correlation) is in fact better than the correlation between RT and total RNA (Fig. 4d), providing robust support for differences across elements in the rate of their converting RNA to RTs, and that a substantial amount of the variability in RT is explained by locus-specific expression.

Notably, the largest cluster (containing 1448 L1 elements that are individually expressed at very low levels) is the highest expressed group in aggregate and is significantly associated with the total RT burden beyond the contributions of the other clusters. Conversely, 20p11.21-1 is assigned a non-significant coefficient despite evidence in this and previous studies that it generates TRTs[11,15], perhaps because the correlation is confounded by heterogeneity of functional and nonfunctional alleles of this locus throughout the population. Two clusters, 9q21.32-2 and 3p22.1-1, are assigned significantly negative coefficients, likely indicating that they are typically expressed in non-RT-permissive contexts.

One limitation of taking total RTs as a response variable is that loci may contribute to RT activity in two ways: they may produce the RNA which becomes the RT substrate, or they can create the machinery which is required for RT activity, without using the substrate originating from its own locus. While the *cis* preference for LINE-1 activity has been demonstrated in the past[37], *trans* activity is possible. *Trans* activating LINE-1 loci may in fact be one possible explanation why there is a discrepancy between the TRT and total RT analyses for some loci and loci clusters. While RNA expression of the source loci is necessary for locus-level observances of TRTs, some loci may vary in their *cis* activity, resulting in differential significance in the latter analysis.

## p53 independently regulates L1 RNA and L1 RT burden

p53, the most frequently mutated gene in cancer[38], is a known regulator of L1 activity[11,18,30–32]. Through its role in the DNA damage response pathway, wildtype (WT) p53 can shut down cells with retrotransposition[29]. WT p53 can also bind genomic L1 promoter loci to repress transcription[33,39,40]. We stratified tumors by p53 mutation status[41,42] and, consistently, found significantly higher L1 RNA expression in the p53 mutant tumors (mean difference of 1.00 between log2-, QC-adjusted TPMs, $p < 10^{-10}$, 2-sided Mann–Whitney $U$ test). This was true when stratifying by tumor type as well, particularly for lung (squamous cell and adenocarcinoma), head and neck, colon, glioblastoma, and liver cancers (Supplementary Fig. 9a, b). L1 RT burden was also higher in p53-mutant tumors (mean difference of 1.57 burden between log2-, QC-adjusted RT counts, $p < 10^{-10}$, 2-sided Mann–Whitney $U$ test), including comparisons within tumor types for lung squamous cell, head and neck, and colon cancers (Supplementary Fig. 9c, d).

The correlation between L1 RT and L1 RNA was stronger among p53-mutant compared to p53-WT tumors ($R = 0.69$, $p < 10^{-10}$ vs. $R = 0.52$, $p < 10^{-10}$, Pearson correlation, Supplementary Fig. 9e–g). Consistent with p53 RT regulation, this suggests that among p53-WT tumors, elevated L1 RNA does not necessarily translate to a higher RT burden, but p53 mutations result in a more direct and proportional relationship between expression and RT. Among p53-WT tumors the correlation is nonetheless strong, as p53 regulates both DNA damage and transcription. On the other hand, the presence of some p53-WT, L1 RNA-high tumors suggest that the p53 regulatory machinery may have been bypassed at both L1 transcription and RT levels.

We sought to test whether associations reported in the past between p53 and L1 RT may have been driven in part by p53 transcriptional regulation of L1 RNA. If WT p53 somatically represses L1 RNA expression, and lower L1 RNA leads to lower L1 RT burden, the apparent regulation of retrotransposition may simply be the downstream result of transcriptional regulation. We found the correlation between p53 and L1 RNA remained significant even when conditioned on L1 RT burden. Further, the correlation between p53 and L1 RT burden remains significant when conditioning on L1 RNA, indicating that p53 independently regulates both stages of the L1 life-cycle (Supplementary Figs. 10 and 11).

To quantitatively evaluate how much of the p53/RT relationship is mediated by p53 impacting L1 RNA upstream of RT, we performed a statistical causal mediation analysis (see "Methods"). The results revealed significant contributions of both p53 directly affecting L1 RT (standardized coefficient $\tau' = 0.19$, $p < 10^{-10}$, OLS) and p53 affecting L1 RT via regulation of L1 RNA (standardized coefficient $\alpha\beta' = 0.18$, $p < 10^{-10}$, OLS; Fig. 7a, b). To test the likelihood of false positive associations in our model, we simulated p53 mutation, L1 RNA expression, and RT burden under two models. In the first, p53 only regulates L1 expression. In the second, p53 does not regulate expression directly, but does lead to cell death in cells with high RT burden (see "Simulating p53 regulation of L1", Supplementary Methods). Neither model was able to recapitulate the significantly mixed effect we see in the TCGA data. These results provide additional support for the proposed dual regulatory role of p53 in restraining L1 retrotransposition[27], with the non-L1 RNA mechanism likely mediated through regulation of genomic instability-associated processes.

We confirmed that these results are robust to the RT caller used (Supplementary Fig. 12a). Where we have sufficient data (at least 20 tumors and variability in p53, for tumor types with a median RT burden above 0), the significance of p53 regulation acting on both L1 RNA and L1 RT is maintained within tumor types (Supplementary Fig. 12b). For individual reference loci with at least 3 TRTs in our dataset ($N = 39$), the extent of p53 regulation of locus-level RNA was somewhat more variable than p53 regulation of locus RT (RNA variance = 0.25, RT variance = 0.14, $p = 0.064$, Levene test), which potentially suggests that some genomic L1 loci may bind WT p53 more strongly and are therefore more impacted by p53 mutations than others, while p53 RT regulation is more element-independent (Supplementary Fig. 13).

We also investigated whether expression of L1 may lead to *TP53* mutations, rather than the reverse, leveraging 670 tumor-adjacent normal samples with RNA-seq and p53 mutation data for the paired tumors. L1 RNA expression in these normal samples may approximate the pre-cancerous conditions of the tumor, and if L1 RNA upregulation can select for, or lead to, p53 mutations, normal tissues with higher L1 RNA expression may be more likely to result in p53 mutant tumors. We first stratified the normal samples by whether the corresponding tumors had p53 mutations and found no significant difference in the L1 RNA levels ($p = 0.94$, 2-sided Mann–Whitney $U$ test, Supplementary Fig. 14a). To rule out spurious associations between tissues with both higher L1 RNA expression and higher frequency of p53 mutations, we also conditioned on tumor type. Here again we found no significant association between normal tissue L1 RNA to tumor p53 mutation status (standardized coefficient = 0.033, $p = 0.49$, OLS, Supplementary Fig. 14b). Of note, the focal area in the pre-cancerous tissue contributing to a p53 mutation may not be the same as the adjacent normal tissue sampled, which would prevent any signal to be detected in this analysis.

## Extension to germline predispositions and other cancer driver genes

Given the recently expanded TCGA dataset, we were able to identify a cohort with germline *TP53* pathogenic/likely pathogenic (P/LP) variants. Due to this increased sample size, we were able to compare tumors from patients with rare germline *TP53* P/LP variants (Li-Fraumeni Syndrome, LFS) against non-LFS tumors, stratified by the same proportion of tumor types. Using permuted resampling of the non-LFS comparison sample sets, we found no difference in either L1 RNA expression or L1 RT burden in LFS compared to non-LFS tumors with *TP53* mutations, suggesting comparable L1 activity throughout tumorigenesis and progression regardless of germline *TP53* status (Fig. 8). The L1 activity in LFS and non-LFS tumors looked the same whether using bootstrapped resampling for the non-LFS dataset or regressing the original dataset against tumor type and comparing the residuals.

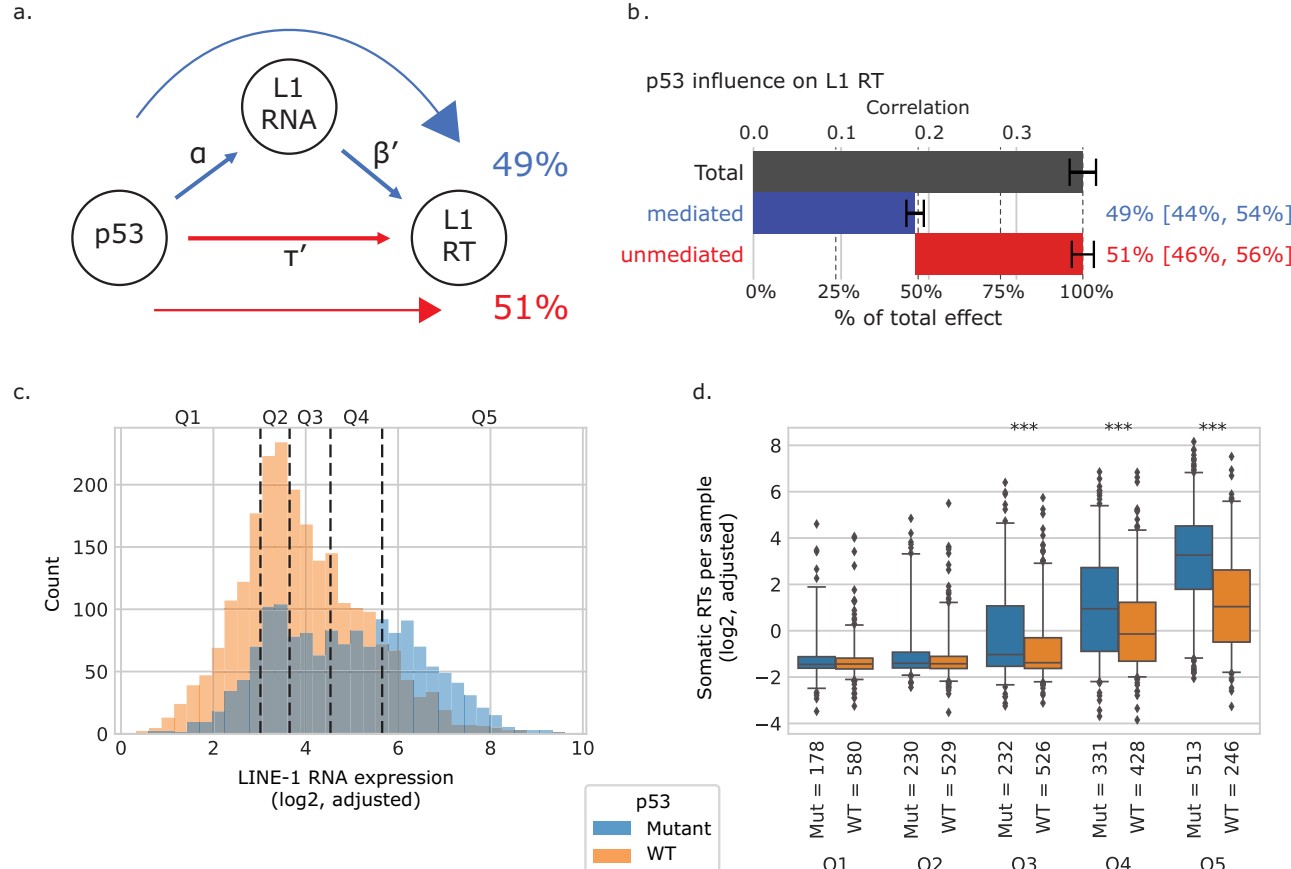

**Fig. 7 | Modeling the regulation of L1 RT by p53 using a mediation model with L1 RNA expression.** *N* = 3793 tumor samples with WGS, RNA, and *TP53* data, excluding tumors with germline *TP53* mutations. **a** Schematic representing the mediation model taking *TP53* mutation as the independent variable, adjusted log2 of L1 RNA expression as the mediating variable, and adjusted log2 of somatic L1 RT burden as the dependent variable. The overall regression of L1 RT versus p53 reflects the sum of the mediated (blue arrow) and unmediated (red arrow) pathways. **b** Breakdown of total influence of p53 on L1 RT by the mediated and unmediated pathways. Total correlation between L1 RT and p53 (gray bar). Contribution of the mediated pathway to the correlation between L1 RT and p53 (blue bar). Contribution of the unmediated pathway to the correlation between L1 RT and p53 (red bar). Top axis, the standardized correlation coefficient between L1 RT and p53. Bottom axis, influence of each pathway expressed as a percent of the total correlation. Error bars indicate 95% confidence intervals determined by bootstrap resampling.

Annotations to the right of the blue and red bars show the estimated percent contributions of each pathway, with 95% confidence intervals in square brackets. **c** Histogram of adjusted log2 L1 RNA expression in p53 mutant and wildtype tumors. Dashed lines divide the dataset into quintiles based on L1 RNA expression. **d** Box plots comparing log2 adjusted L1 RT burden in p53 mutant versus WT tumors, stratified by L1 RNA quintiles from (**c**). Center line indicates median. Box indicates interquartile range. Points more than 1.5× IQR away from the box are shown as individual outliers. Bonferroni adjusted significance level of two-sided Mann–Whitney U test comparing p53 mutant tumors to p53 WT tumors within a quintile: *$p < 0.01$, **$p < 0.001$, ***$p < 0.0001$. Exact *p*-values and effect sizes for each quintile are as follows. Q1: 1.0, 0.02; Q2: 0.43, 0.08; Q3: $5.86 \times 10^{-5}$, 0.20; Q4: $2.6 \times 10^{-7}$, 0.23; Q5: $1.62 \times 10^{-27}$, 0.49. Ns as shown in figure. c-d) Blue bars/boxes indicate p53 mutant tumors. Orange bars/boxes indicate p53 WT tumors.

Thus far, we have shown that the variability in L1 RT burden can be explained to a large extent by locus-specific L1 RNA expression, tumor type, and *TP53* mutation. In particular, *TP53* mutation partitions L1-active tumors from L1-repressed tumors. However, additional regulatory factors may also exist. To this end, we took cBioPortal annotations for the 82 most frequently mutated genes in the TCGA dataset and looked for those that partition L1 RNA or L1 RT within the *TP53* mutant or WT cohorts (Supplementary Fig. 15).

Following multiple hypothesis correction, *IDH1* and *ATRX* mutants were found to associate with significantly different L1 expression when *TP53* is also mutated. In both cases, L1 RNA is comparable within the *TP53* WT cohort, but in the *TP53* mutant cohort, L1 RNA was significantly repressed in *IDH1* mutant and *ATRX* mutant tumors. These two genes have the same significant impact on L1 RT in the *TP53* mutant tumors. Within *TP53* wildtype tumors, mutations in *RELN*, *PTPRT*, and *CDKN1A* are all associated with higher L1 RT burden. In *TP53* mutant tumors, in addition to wildtype *IDH1* and *ATRX*, mutations in *TTN*, *LRP1B*, *CDKN2A*, *SYNE1*, *PTPRD*, *NAV3*, *RELN*, and *PKHD1* are all significantly associated with higher L1 RT burden (two-sided

Mann–Whitney *U* test Bonferroni-adjusted *p*-value < 0.01) (Supplementary Fig. 15).

## Discussion

To our knowledge, this work includes the first large-scale pan-cancer analysis of the expression levels of active L1 mRNA, with a focus on the relationship between expression level and retro-transposition (RT) activity, making our assessment the most comprehensive to date. We present a new RT caller, TotalReCall, which together with the previously published tool xTea, can improve specificity while maintaining the sensitivity gains of methods that prioritize clipped reads over discordant reads. Both callers had close agreement on which insertions contain inversions, which we found occur at a higher rate in tumors compared to germline[18]. As a byproduct of our search for tumor-specific somatic L1 insertions, we also characterized the distribution of a conservatively-defined subset of non-reference germline insertions in this population, but due to the intentionally non-exhaustive nature of these, our findings should be interpreted with caution (see "Non-reference L1

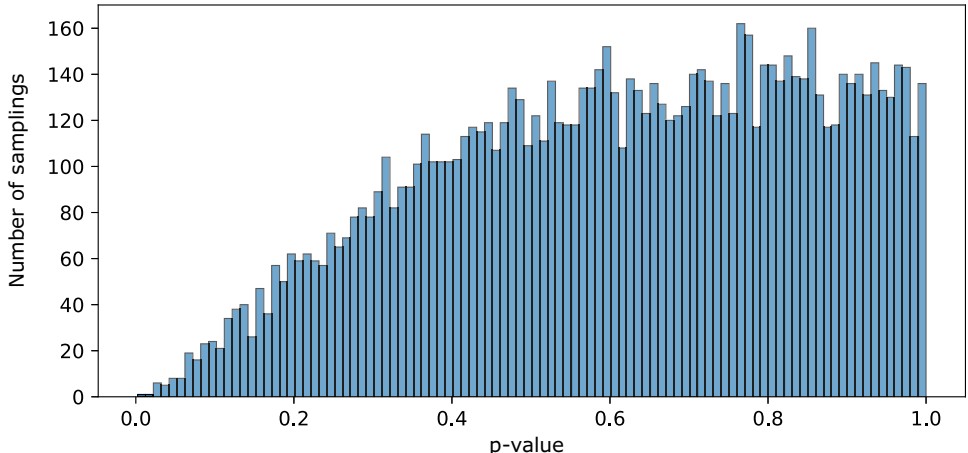

**Fig. 8 | L1 activity in cancer is not affected by germline *TP53* mutations.**
**a** Comparing L1 RT burden (top) and RNA expression (bottom) in tumors from patients with germline p53 mutations ($N = 13$, RT burden; $N = 22$, RNA expression) compared with bootstrap resampling controls of patients with somatic p53 mutations ($N = 10,000$ resamplings). *p*-Values are calculated from a two-sided *t*-test of bootstrapped values compared with true LFS samples. **b** 95% confidence intervals of the means of bootstrap resamplings, with the red dotted line representing the value for the true LFS sample set.

insertions present in both case and control samples", Supplementary Methods).

The sensitivity of our analyses informs understanding of L1 behavior in different contexts. For instance, while L1 RNA expression generally correlates with RT activity, as previously observed in a study of gastric cancers[18], we found indication-specific signals: in liver and kidney cancers, there are fewer RTs than would be expected given the RNA expression level (Fig. 4c). In prostate cancers, there is significant L1 RT activity despite L1 RNA expression being comparable between prostate cancer and normal prostate.

Differences in RNA expression versus RT activity level by indication and tissue type also have the potential to be clinically meaningful, particularly as our understanding of the various mechanisms by which the L1 life cycle interfaces with human health deepens. A previous study quantifying L1 RNA and ORF1p in a limited subset of cancers[32] suggested the oncogenicity of L1 derepression is greater than what can be attributed to the completed cycles of retrotransposition (e.g., disruption of non-L1 gene expression[32]). Beyond RT activity, an abundance of L1 transcripts can play a role in heterochromatin erosion[43], and the presence of cytosolic L1 RNA:DNA

hybrids resulting from reverse transcription could lead to an inflammation response that can alter the tumor microenvironment by activating the cGAS innate immune response, which in turn activates the interferon pathway[44].

Expanding significantly upon previous in vitro work[24] by assessing all active L1 loci in the human genome, we find a wide range of efficiencies across L1 loci, including differential behavior across cancer types. For example, while the expression levels of 22q12.1 are similar across several indications, including cervical, bladder, and ovarian cancers, its RT activity appears much higher in cervical cancer. By necessity, we rely heavily on using TRT events as a proxy for RT events. As most RT events do not result in transductions, we may not have the data to accurately quantify efficiencies of many L1 loci. The consistency between transduction-dependent efficiency estimates presented here and earlier transduction-independent in vitro measurements[24] provide strong support for our results in spite of this limitation. Still, studies have demonstrated differential RT-competency across loci for over two decades[20,24,25,45], and it is widely accepted that some loci are "hotter" than others in terms of either observed RTs or RNA[11,24–26]. It is often assumed, however, that RT rates

are proportional to RNA regardless of locus (provided that a locus is RT-competent)[14,21]. Here we provide strong evidence across cancers that this is not the case.

This is of particular interest because the reference alleles of the RT-competent L1s vary from the consensus sequence by only a few amino acids. Thus, even small differences may impact RT efficiency. Such differences could affect, for example, the kinetics of ORF2p reverse transcriptase enzymatic activity, altered translational efficiency of L1 RNA, binding efficiency of ORF1p to L1 RNA, cellular co-localization of ORF1p with ORF2p, ability of ORF2p to localize to the nucleus, and/or the rate of genomic nicking by endonuclease. The significant heterogeneity across loci revealed in this study speaks to the value in considering individual L1 elements separately as potential clinical targets and biomarkers.

We have focused our analyses primarily on L1 loci present in the reference genome, though we found a significant number of TRTs originating from polymorphic loci, consistent with literature reporting that these latter elements may be among the most active[20,25,26]. Their existence is also of broad interest as we build our understanding of the role of L1 in human health and disease. A deeper dive into the landscape of L1 polymorphism is warranted but is beyond the scope of this study. Nonetheless, the results presented here are likely to be robust, given the high predictability of L1 expression on RT burden even absent any additional covariates suggests significant variables are not missing (Supplementary Fig. 8).

Finally, we demonstrated how large-scale, multi-modal modeling can disentangle the interplay between canonical driver events in tumor evolution and retrotransposon activity. Many studies have noted that the role of p53 as a general regulator of cell cycle control, apoptosis, and senescence is insufficient to explain the extent of tumor suppression by p53[46-48]. Here we provide evidence in primary human tumors for a dual role of p53: (a) in response to retrotransposition, p53 may activate various cellular stress response pathways that ultimately prevent the growth of cells harboring L1 RT events[42,49-51]; (b) p53 may modulate L1 activity directly, for example, by acting on the L1 promoter to down-regulate L1 RNA expression.

Our model assumes that L1 expression and RT activity, and p53 mutation, can be integrated, although they operate on different time scales. Thus, any correlations, or lack thereof, should be interpreted with some care. Nevertheless, our analysis suggests that p53 derepression of L1 through direct binding of the genomic L1 5' UTR (previously demonstrated in vitro[33,39,40]) is likely to be somatically active across tissues and significantly mutagenically disrupted in vivo in cancers.

Beyond *TP53*, we additionally nominated 12 genes whose mutational states are significantly associated with L1 activity; further experiments are needed to determine which of these associations may reflect direct regulatory mechanisms. From a clinical perspective, we showed RT activity in tumors from patients with LFS, suggesting future therapeutic trials targeting L1 should include individuals with LFS, who are at very high risk of developing tumors. While additional data are needed, as these patients undergo early detection and screening, it is further compelling to consider the incorporation of early cancer interception with such a L1-targeting agent.

We also found L1 RT burden to be significantly associated with a worse clinical prognosis in pooled tumor samples (Supplementary Fig. 16). Interestingly, when split by tumor type, this association is no longer significant, suggesting a strong correlation between aggressive cancers and those with high L1 burden. We speculate that L1 and transposable elements are likely central to the evolution of the DNA damage response, in a way that extends far beyond p53.

## Methods

The study complies with all relevant ethical regulations. The data used in the study (TCGA and the GIAB) do not require IRB approval.

### Pan-cancer dataset (TCGA)

Whole-genome sequencing data (releases 37.0 and 38.0, 2023) were accessed from the Genomics Data Commons (GDC) cloud storage, using the GA4GH standard Data Repository Service (DRS) for URI resolution. RNA-Seq data were downloaded from the GDC directly (https://portal.gdc.cancer.gov/repository) as hg38-aligned BAM files using the GDC Data Transfer Tool (https://docs.gdc.cancer.gov/Data_Transfer_Tool). All subsequently described processing of these data was performed within the Terra.bio cloud data platform (https://app.terra.bio/).

### TotalReCall method for detection of L1 retrotranspositions

Retrotransposon insertions can be detected from short-read sequencing from fragments which cross the breakpoint. Depending on the start and end of the fragment, this may result in a discordant read pair (when the two paired reads from both ends of the fragment are not properly mapped as a pair) and/or split ("clipped" in mapping jargon) reads (when one of the reads spans the breakpoint) (Fig. 1). In earlier TCGA releases, the length of the reads was 50 bp or less, which typically resulted in the inability to accurately map split reads that crossed RT insertion breakpoints. As a result, methods for retrotransposition detection (e.g., MELT[17] and TraFiC-mem[11]) used discordant read pairs as their primary signal. As 150 bp paired-end reads have become the *de facto* standard for whole-genome sequencing, the importance of clipped reads has increased. For example, xTea[19], a newer method for retrotransposon detection, uses both discordant read pairs and clipped reads. In this paper, we developed the "TotalReCall" approach, which explicitly accounts for different modes of the retrotransposition process (Fig. 1a–d). Similarly to xTea, it relies on the two key signals outlined above. In detail (Fig. 1g), when aligning sequencing reads to the reference genome, TotalReCall looks for: (i) reads that span the retrotransposon insertion site breakpoint and result in chimeric reads that contain portions of both the insertion site sequence from the human genome as well as the inserted L1 sequence, leading to "soft clipped" alignments; and (ii) paired-end reads that arise from fragments spanning the inserted L1 sequence, which often have one read mapping near the insertion site but the other read mapping to one of the many L1 sequences elsewhere in the genome (even to a different chromosome or a distant site on the same chromosome), resulting in "discordant read pairs".

While xTea uses clipped reads to determine the breakpoints, TotalReCall additionally uses clipped reads (rather than discordant read pairs) to infer the transposon length and inversion status (Fig. 1). Indeed, a split 150 bp read can be mapped reliably (there are some subtleties related to mappability and low complexity regions), such that the clipped sequence is long enough to be reliably mapped to the transposon consensus (Supplementary Fig. 17). In addition, TotalReCall utilizes (typically hard-clipped) supplementary alignments that infer the sequence of the clipped part using the matching primary alignments (see "L1 retrotransposition detection from short-read whole genome sequencing data", Supplementary Methods for more details).

### Validation using Genome in a Bottle (GIAB) dataset

Alignments to hg19 and indices (BAM and BAI files, respectively) for Oxford Nanopore (ONT) long reads and Illumina reads for the Genome in a Bottle benchmarking project[35] were downloaded from the NCBI archive. A complete list of access links can be found in Supplementary Data 4. Illumina reads (300x coverage, 150 bp paired end reads) were sorted by name and reverted to FASTQ and then aligned to hg19 (in order to be able to run TraFiC-mem, which is hardcoded to use hg19) using bwa. Duplicates were marked using samtools. Then the resulting alignments were randomly downsampled to 80× and 35× (to match the TCGA whole-genome sequencing protocol). We used reads for the trio of Ashkenazi Jewish ancestry (HG002/HG003/HG004) because all the

family members were sequenced to the same depth. Also, one sample (HG005, child) from the trio of Han Chinese ancestry had a very low quality (high fraction of discordant read pairs); as a result, we skipped these three samples. We ran TotalReCall, xTea[19] and TraFiC-mem[11] using different members of the trio as "case" (downsampled to 80x) and "control" (downsampled to 35×), resulting in 6 pairwise comparisons. We checked for supporting insertions in the ONT reads to verify the correctness of the calls (see "Validating retrotransposition detection by TotalReCall, xTea, and TraFiC-mem using long reads", Supplementary Methods for details). All the calls in the intersection set of the TotalReCall and xTea calls were supported by the long reads, and the total number of calls made by both callers was highly correlated across samples (Supplementary Figs. 18 and 19). This motivated us to use the intersection of TotalReCall and xTea calls for TCGA. A small fraction of these calls had conflicting inversion status, which were then manually verified using the inserted sequences from the Nanopore reads and running BLASTn against the L1 consensus sequence. This confirmed the correctness of the TotalReCall inferences of inversions for the L1 retrotranspositions (Supplementary Data 5).

**Short-read whole-genome sequencing (WGS) data from TCGA**
**Quality control data filtering.** Variability of sequencing library quality across TCGA WGS data can obscure the differentiable signal of biological relationships. With that in mind, we filtered the entire WGS dataset (15,034 samples from 6260 patients, as of release 38.0) to only those samples with alignment files from GDC Data Release 37.0 or 38.0 stored in NCI Genomic Data Commons (10,312 samples from 5148 patients). We removed 14 non-primary tumor samples and divided the remaining samples into 5161 tumor-normal pairs (which includes some redundant pairs per patient, removed later). To account for confounding differences between the libraries and better capture biological relationships, the quality of each WGS library was evaluated using a modified Picard[52] tool to quantify the total number of reads, average base quality, average read length, and fractions of reads with split or discordant alignments (the two sources of information on which TotalReCall relies). All reads that did not have a matched pair, that were marked as duplicates, that had mapping quality equal to zero, or whose base and quality score strings were inconsistent (i.e., of differing lengths) were removed from the analysis. From the remaining reads, for each sample, we calculated: (1) the fraction of chimeric reads, (2) the fraction of overall clipped bases, (3) the average read length, (4) the average base quality, and (5) the total coverage. Note that a chimeric read is defined per Picard/GATK v4.1.8.1 as a read pair aligned in an unexpected orientation or significantly further apart than expected (with maximum insert size set to 100,000). We removed all paired samples with a chimeric read fraction in the tumor or normal sample greater than 2%, tumor samples sequenced below 50x depth, and normal samples sequenced below 20× depth, resulting in a total of 4688 tumor-normal pairs from 4669 patients. To de-duplicate individual patients with multiple primary tumor or normal samples, only the samples with the longest read lengths per patient were retained. TotalReCall depends on signal from split-read alignments (i.e., clipped reads) to nominate candidate insertion sites, which will only be aligned as expected by bwa mem for read lengths of at least 70 bp. In GDC Data Releases 37.0 and 38.0, all libraries were sequenced with at least 100 bp reads; for improved consistency, we selected the QC-passing primary tumor-normal pair for each patient with the longest read lengths, which resulted in a dataset of 4669 tumor-normal pairs across 31 tumor types in which all samples had been sequenced with 150 bp reads. Due to their importance, the five metrics for each tumor and paired normal sample calculated here were also used to adjust the retrotransposition count estimates for each tumor sample, as shown in Supplementary Fig. 20 and described below. The final dataset of 4669 tumor-normal pairs can be found in Supplementary Data 6. Values for all QC metrics (10 per pair) are included in Supplementary Data 3.

**Identification of somatic L1 retrotransposition**
**TotalReCall.** TotalReCall was run on 4669 TCGA tumor-normal pairs using all default parameters through the Terra.bio cloud data platform. See "Code availability" section below for information on running the workflow, which is written in WDL.

**xTea.** The xTea (v0.1.7) short read module was run on the 4669 TCGA samples for somatic L1 insertion identification. xTea first calculates the average depth and then automatically adjusts the parameters based on the calculated depth. In addition, we set the tumor purity to 0.45 ("--purity 0.45"). When parsing transductions from the resulting calls, we further grouped the identified L1 transductions whose traced source elements are within 1000 bp, and we disregarded transductions whose source fell within sequence that could multimap to L1 sequence (the RT call was kept, but it was treated as non-transduction-bearing).

**Intersecting TotalReCall with xTea.** Calls from both call sets were removed if the insertion did not occur on one of the 24 canonical reference chromosomes. "Orphan" calls from xTea (designated in the TD_SRC field) were also removed. All calls identified in the same sample with coordinates (CHROM and POS) within 50 bp from each other were considered shared. Nearly 96% of shared calls shared the exact coordinates identified by both callers, and >99% were <10 bp apart. All calls from the intersection set can be found in Supplementary Data 7.

**Inversion rate across cancer subtypes.** The intersection of xTea and TotalReCall retrotransposition calls as listed in Supplementary Data 7 was used to calculate the inversion rate across somatic RTs. Only indications with more than 500 total LINE-1 RT events called in this intersecting list are shown. The error bars represent the 99% Clopper–Pearson confidence interval computed using the binom R package.

**RNA sequencing**
**Reprocessing of alignment files.** Public RNA-seq data from TCGA were reverted to unaligned FASTQ format using GATK v4.1.8.1 tools RevertSam (with options "--SORT_ORDER "queryname" --VALIDATION_STRINGENCY SILENT") and SamToFastq (with options "INTERLEAVE=false INCLUDE_NON_PF_READS=true"). Any reads that were not paired were dropped. The paired-end FASTQ files were then aligned to the hg38 human reference genome using STAR v2.7.9a.

**Quality control data filtering.** Alignment files for a total of 10,904 RNA-seq samples from 10,089 individuals (10,174 tumor samples and 730 normal samples) were downloaded from GDC as described above. Twenty-eight of these (all tumor samples) could not be reverted to FASTQ for realignment due to the presence of unpaired reads or otherwise corrupted downloaded alignment files. The remaining 10,876 samples were realigned as described above. 634 of these (623 tumor samples and 11 normal samples) were sequenced with single-end reads, and therefore removed from our dataset. 72 additional tumor samples were removed from our dataset for containing a strand-specific sequencing library. Finally, we filtered out any metastatic, recurrent, or new primary tumor samples and deduplicated the patient samples in the dataset to include no more than one primary tumor and one normal sample per patient, resulting in a dataset of 8998 tumor samples and 719 normal samples from 9071 individuals across 32 tumor types. These samples are listed in Supplementary Data 8.

**L1 RNA quantification.** L1 RNA expression was quantified in the 8998 tumor and 719 normal RNA-seq samples using L1EM[28], which formalizes a framework for quantification of expression that is based on the mechanisms of active transcription of L1 elements. Read counts for

proper expression at all loci were converted to transcripts per million (TPM) of active L1 expression based on transcriptome-wide gene counts assigned by STAR. Read counts for run-on expression ("3prunon" in L1EM) were also converted to TPM, using the same "per million" denominator per sample as active expression.

**Comparing RT calls to PCAWG.** RT calls identified in the PCAWG dataset were obtained from Supplementary Data 2 of Rodriguez-Martin et al.[11]. To match samples shared between the PCAWG and TCGA datasets, a sample spreadsheet was downloaded from the ICGC Data Portal (https://dcc.icgc.org/api/v1/download?fn=/PCAWG/donors_and_biospecimens/pcawg_sample_sheet.tsv, accessed 31 July 2023) and matched to the TCGA dataset using the submitter specimen id. This ensured that only one sample per patient was included in the comparison, consistent with our dataset. The calls from PCAWG were converted from hg19 to hg38 coordinates using the liftover package in Python. The intersection was then performed in the same manner as the intersection between xTea and TotalReCall described above. Calls from the same sample on the same chromosome were considered "shared" if the leftmost coordinates of the target site as identified in both call sets were within 50 bp of each other.

**Categorizing tumors as p53 wild type (WT) or mutant**
Categorical designations of p53 alteration were obtained through the public repository cBioPortal (www.cbioportal.org, accessed 23 February 2023) by querying "TCGA PanCancer Atlas Studies" (which includes 10,967 samples from 10,953 patients in 32 studies) for all alterations in "TP53". A sample will be marked as "altered" in *TP53* if any non-synonymous mutations, amplification, deep deletion, or structural variants have been identified in that sample. Shallow deletions or low-level copy number gains of the gene do not contribute to classification of a sample as altered. The altered annotations were used to divide tumor samples into p53 mutant and wild type (WT) categories to test significance with Mann-Whitney U tests as shown in Supplementary Fig. 9, and to demonstrate robustness of the mediation model between p53, L1 RNA, and L1 RT burden as shown in Supplementary Fig. 10.

**Categorizing tumors as wild type (WT) or mutant for additional genes**
Categorical designations of alteration status for the 82 most frequently mutated genes in TCGA were obtained through cBioPortal (www.cbioportal.org, accessed 23 February 2023) using the same query and categorization as above. The sample-level annotations for the dataset used in this study can be found in Supplementary Data 3. The complete list of genes and results of stratified tests for L1 RNA and RT association can be found in Supplementary Data 9.

**Identifying subsets of L1 elements**
This and previous studies[26] have identified transductions linked to source elements that do not have intact ORF1 and/or ORF2 domains in the reference genome. We therefore use all L1HS and L1PA2 loci that are annotated as "category 1" (meaning the 5′ end of the element is sufficiently intact to function as a promoter) by L1EM[28] throughout our analyses. We also make use of a subset of 121 L1HS and L1PA2 elements in Fig. 3 and Supplementary Figs. 3 and 5, which is based on the combination of source elements of transductions identified in this study and elements with in vitro evidence of retrotransposition as previously annotated in Supplementary Data 23 of Ebert et al.[26] (which are also present as category 1 in L1EM). Locus Xq21.1-3 is not included in the set of RT-competent elements due to inconsistent coordinate annotation in Ebert et al.[26]. When calculating efficiencies, we additionally included all loci that had been profiled in vitro[24], by first extracting the sequences of L1 elements within the cytoband

sequences provided by accession numbers in Supplementary Data 6 of Brouha et al.[24] and then realigning those sequences to the hg38 reference genome, resulting in 156 total L1 loci, 76 of which were shared with Brouha et al.[24] and used in Fig. 6b. Locus 22q13.32 is included in the set of 156 elements but not in the comparison set of 76 due to being included in the Brouha et al. study but unsuccessfully assayed. The loci included in each subset are reflected here in Supplementary Data 10.

**Adjusting RNA and RT estimates based on sequencing metrics**
Intronic rate of RNA-seq, which has previously been shown to be a key confound for L1 expression[53], was calculated using RNA-SeQC v2[54]. A linear regression between the L1 RNA estimates and the intronic rate was then performed in Python v3.7.12 using statsmodels v0.13.0 OLS to perform an ordinary least squares regression. The residuals from this model were then used as the adjusted L1 RNA estimates (Supplementary Fig. 20a). Tumor and normal RNA-seq were fitted in the same model.

Similarly, quality metrics for the WGS samples were calculated as described above. An ordinary least squares model was fitted to the tumor-specific retrotransposition counts as a function of the average read length, depth of coverage, average base quality, chimeric read fraction, and clipped base fraction in both the tumor sample and the paired normal sample. The residuals from this model were then used as the adjusted L1 RT estimates (Supplementary Fig. 20b).

**Clustering L1 loci based on RNA and TRT across tumor types**
To condense the 1483 L1HS and L1PA2 loci into 13 clusters of loci with similar activity, we first considered the subset of our dataset with both WGS and RNA-seq data ($N = 3879$ tumors across 29 tumor types). Using the QC-adjusted log2 values for both RNA and TRT measurements, we calculated the mean value per tumor type at every locus. The histograms shown in Fig. 5a, b are based on the mean across tumor types of these mean-per-tumor type values. We then constructed two dendrograms, one for the RNA means and one for TRT means, using the clustermap function of the seaborn library in Python, and defined clusters within each dendrogram using the fcluster function from the scipy module cluster.hierarchy.

Finally, we annotated each locus based on whether the mean TRT value across tumor types fell within the lowest histogram bin (fewer than 0.0018 log2, QC-adjusted TRTs per sample). Loci that shared all three annotations (2 cluster assignments from the RNA and TRT dendrograms, and the TRT histogram-based annotation) were assigned to the same cluster, resulting in the final set of 13 clusters used in Fig. 5 and Supplementary Fig. 8. Additional details can be found in *Locus clustering by RNA expression and RT events* in the Supplementary Methods. The mean values across all loci within each cluster are shown in the heatmaps. Clusters are sorted based on high-to-low mean across tumor types of TRT values. Tumor types are sorted based on high-to-low sum of mean locus RNA. For the 121 RT-competent loci (identified as described above), we repeated the same procedure (Supplementary Fig. 5).

**Modeling locus efficiency**
Statistical analysis was performed in Python v3.7.12 using statsmodels v0.13.0 OLS to evaluate all efficiency models. A notebook including the python code to fit all models described below is included in the github repository associated with this study, described in the "Code availability" section below.

**Efficiency model.** Efficiency was calculated for 156 L1 loci, identified as described above. The subset of our dataset with WGS, RNA-seq, and p53 mutation annotations was used ($N = 3820$ tumors). Log2, QC-adjusted values were used for both locus RNA and TRT. A single

ordinary least squares regression was fitted for the model

$$locus_i TRT \sim efficiency_i \cdot locus_i RNA + tumor\ type + p53 + const$$

treating tumor type as a categorical variable, such that every tumor type and p53 mutation status were assigned the same coefficients within every locus model. The input data was formatted such that each observation represents a single tumor and locus, with $3820 \times 156$ total observations.

**Background model of efficiency.** In the above model, every locus will be fitted with a coefficient to quantify the contribution of locus RNA to locus TRTs above and beyond what is expected across any locus based on the tumor type and mutational status of p53. To test whether these coefficients differed from each other more than would be expected due to noise, we generated 1000 resampled variations of the input data to the OLS model, where the locus associated with each observation was randomly permuted. For these resampled inputs, we used only data from 48 L1 loci with TRTs in at least 2 different samples in the dataset of 3820 tumors. Doing so enabled greater variability in the simulated background efficiencies, and the apparent significance of true efficiencies would have been even higher compared to a background of all 156 loci. We thus generated a total of 48,000 resampled efficiency estimates, used to define empirical significance of each true efficiency estimate, and the variance of efficiencies within each of the 1000 resamplings was used to define the empirical significance of the true efficiency variance.

**Categorizing loci based on background model.** All 156 loci with fitted efficiencies were compared against the background model of 48,000 permutation resampled efficiencies to calculate a one-sided empirical significance of an efficiency as high or as low as the true fitted value. These *p*-values were multiple hypothesis corrected (including all 156 loci as possible hypotheses), and corrected significance was set at a threshold of 0.05. Coefficients that did not meet the criteria for significance but fell outside the interquartile range of the background model were categorized as "slightly high" (above the 75th percentile of the background model) or "slightly low" (below the 25th percentile of the background model). All other loci were categorized as "typical". All 156 loci were assigned efficiencies and reported in Supplementary Data 10, but only the 48 loci included in the background model are shown in Supplementary Fig. 6.

**Total RT ~ locus RNA model.** Individual ordinary least squares regressions were fitted for 121 L1 loci with evidence of transductions, identified as described above. For these loci, coefficients were fitted for the model

$$Total\ RT\ burden \sim coef \cdot locus\ RNA + const$$

based on input data from 3879 tumors with both WGS and RNA-seq data. "Total RT burden" is used here to differentiate from the locus-specific TRT burden used above and is equivalent to "RT burden" used elsewhere in the manuscript, e.g., as seen in Fig. 2. Log2, QC-adjusted values were used for both RT and locus RNA. The coefficients for each of the 121 models are recorded in Supplementary Data 10.

**Background model of locus correlation with total RT.** To generate the background model, 1000 resampled variations of the model input data were generated by randomly resampling with replacement observations across any sample and locus, and then permuting the locus assignment of locus RNA. For each input resampling, 121 separate regression models were fitted, resulting in 121,000 total background estimates for the coefficients of locus RNA correlation with total RT. The variance of coefficients within each of the 1000 resamplings was used to determine the empirical significance of the variance of the true coefficients.

**Categorizing loci based on background model.** As above, all 121 loci with fitted coefficients were compared against the background model of 121,000 resampling-based coefficients to calculate a multiple-hypothesis corrected one-sided empirical significance. Categories were assigned based on a significance threshold of 0.05 and values within or without of the interquartile range of the background distribution as described above.

**Total RT ~ Σ cluster RNA model.** The clusters defined in Fig. 5 and described above were used as inputs to the model

$$Total\ RT\ burden \sim \sum_{clusters} c_i \cdot cluster_i RNA + const$$

where $cluster_i RNA$ is the sum of locus RNA from all loci assigned to a given cluster within a single sample. Log2, QC-adjusted values are used for RNA and RT. As before, "Total RT burden" is used here to indicate the dependent variable of this model is not locus-specific and is equivalent to "RT burden" used elsewhere. All tumors with WGS and RNA-seq data ($N = 3879$) were used as inputs to this model.

## Identifying the Li-Fraumeni Syndrome cohort
To create a TCGA-LFS cohort, carriers of putative germline *TP53* P/LP variants were identified from the TCGA pan-cancer analysis[55]. Tumor and matched normal DNA BAM files were downloaded from the Genomic Data Commons (GDC) using a National Center for Biotechnology Information Genotypes and Phenotypes Database (NCBI dbGaP phs000178) approved protocol (#21931) and underwent quality control to ensure the *TP53* variant was found at the expected heterozygous frequency in the normal BAM file; variant classification was performed using the American College of Medical Genetics (ACMG) specific guidelines[56]. Annotations indicating which samples in this study belong to the TCGA-LFS cohort are provided in Supplementary Data 3.

## Mediation model
Statistical analysis was performed in Python v3.7.12 using statsmodels v0.13.0 OLS to evaluate the mediation model. For the 3847 *TP53* mutated tumor samples without germline *TP53* mutations for which we had WGS, RNA-seq, and *TP53* mutation data available, five linear regressions were fitted (see Supplementary Fig. 10), using the log2, QC-adjusted values for RT and RNA:

$$RT \sim \tau \cdot p53\ mutation\ score + c_1 \tag{1}$$

$$RT \sim \beta \cdot RNA + c_2 \tag{2}$$

$$RNA \sim \alpha \cdot p53\ mutation\ score + c_3 \tag{3}$$

$$RT \sim \beta' \cdot RNA + \tau' \cdot p53\ mutation\ score + c_4 \tag{4}$$

$$RNA \sim \beta^* \cdot RT + \alpha^* \cdot p53\ mutation\ score + c_5 \tag{5}$$

Note that only Eqs. (3) and (4) are necessary for quantifying the magnitude and significance of the mediation. Equation (1) is used to normalize the mediated and unmediated effects to the total effect of p53 on L1 RT burden. Equation (2) is included here to evaluate the correlation between L1 RNA and RT burden alone, without considering p53. Equation (5) (Supplementary Fig. 10b) was evaluated to confirm that p53 has a significant effect on L1 RNA even when controlling for L1

RT burden. The significance of this model fit was further explored using simulations to model an empirical distribution for the covariate-controlled contribution of p53 mutation to both RNA and RT, described further in the Supplementary Methods. In each model, $c_i$ incorporates both the intercept and the error terms. The fitted coefficient values were then standardized based on the estimated standard deviations for each variable[57]. The ratio of $\tau'$ to $\tau$ gives the estimate for the percentage of the total impact of p53 on L1 RT burden that is not mediated by L1 RNA. The product of coefficients $\alpha$ and $\beta'$ (which is equivalent to $\tau-\tau'$) gives a coefficient for the effect of p53 on L1 RT burden via L1 RNA, and similarly the ratio of $\alpha\beta'$ to $\tau$ gives the estimate for the percentage of the total impact of p53 on L1 RT burden that is mediated by L1 RNA. Standardized coefficients are calculated by multiplying the original coefficient by the ratio of the standard deviation of the dependent variable to the standard deviation of the fitted variable. 95% confidence intervals for coefficient values are provided by the OLS outputs. 95% confidence intervals for mediated fractions are provided by concurrently running statsmodels Mediation with Eqs. (4) and (3) as inputs.

### Comparing L1 RNA and RT in LFS and non-LFS, *TP53* mutant tumors

Tumors from individuals with germline *TP53* P/LP variants (Li-Fraumeni Syndrome) were compared against all other tumors, controlling for tumor type composition in various ways. First, linear regression models were fitted for log2 RT burden or log2 RNA as a function of tumor type (treated as a categorical variable). We generated 10,000 samplings (with replacement) of non-LFS, *TP53* mutant tumor samples, matched by indication, to compare RT burden and RNA expression against the true LFS cohort ($N = 13$ for the WGS data, and $N = 22$ for the RNA-seq tumor data). The distributions of log2 RT and log2 RNA were compared between the true LFS datasets and the resampled non-LFS datasets. Significance of the differences between log2 RT and log2 RNA values between subsamplings and the true LFS data set was evaluated with a two-sided *t*-test.

### Causal mediation models of L1 RNA influencing *TP53* mutation

For 670 non-LFS normal RNA-seq samples for which there is a tumor *TP53* mutation annotation (obtained from cBioPortal as above) for the same patient, log2 QC-adjusted RNA values were compared in the patients with somatic *TP53* mutations to patients without somatic *TP53* mutations and tested with a two-sided Mann–Whitney *U* test, as implemented in the scipy.stats module. For those same samples, an ordinary least squares regression was fitted for the model

$$tumor\ p53\ mutation\ status \sim normal\ RNA + tumor\ type + const$$

where tumor type is treated as a categorical variable, such that each tumor type is assigned its own coefficient reflecting the likelihood of *TP53* mutation in the tumors. Coefficients for each variable are standardized by multiplying the original coefficient by the ratio of the standard deviation of the dependent variable (binary value representing tumor *TP53* mutation) to the standard deviation of the corresponding variable[57].

### Testing significance of gene mutations

For the 82 most commonly mutated genes in TCGA (as recorded by cBioPortal), a series of Mann-Whitney *U* tests was performed. First, within the dataset of 3820 tumors with WGS, RNA-seq, and cBioPortal mutation annotations, two-sided Mann-Whitney U tests compared the log2, QC-corrected L1 RNA and RT burden in samples that were wild-type vs. mutated in each gene, resulting in 164 comparisons. Next, given the abundance of *TP53* mutations and the known regulatory association between p53 and L1, we stratified the remaining 81 genes into subsets of p53 WT tumors ($N = 2329$) and p53 mutant tumors

($N = 1491$) and again compared L1 RNA and RT burden for tumors mutated or wildtype in each gene, resulting in 4 two-sided Mann-Whitney *U* tests per gene. We recorded the mean and median values for RNA and RT within every subset of p53 mutant or wildtype, and gene mutant or wildtype groups of tumors. Any comparisons that were significant below a threshold of 0.01 following multiple hypothesis correction and also had an effect size (the magnitude of the difference between the median RNA or RT values in the mutated and wildtype subsets) at least as large as the overall standard deviation of RNA or RT values throughout our dataset were included in Supplementary Fig. 15. The results of all 6 comparison tests for all 82 genes are included in Supplementary Data 9.

### Reporting summary

Further information on research design is available in the Nature Portfolio Reporting Summary linked to this article.

## Data availability

All raw data have been obtained from publicly available datasets, with references cited herein. Specific Genome in a Bottle sequencing data download links are provided in Supplementary Data 4. Processed data generated in this analysis, in addition to TCGA sequencing data access information, are provided as Supplementary Data. Source data for figures are provided. Pseudo-germline calls are available as controlled access data at the database of Genotypes and Phenotypes (dbGAP) as controlled access data under the project phs003888 (https://www.ncbi.nlm.nih.gov/gap/). Data access requests will be handled by dbGAP directly. For specific concerns, contact: NCIDAC@mail.nih.gov. Source data are provided with this paper.

## Code availability

TotalReCall code for L1 retrotransposition detection from short-read whole-genome sequencing data is publicly available on GitHub (https://github.com/Rome-Tx/totalrecall, https://doi.org/10.5281/zenodo.14553438). The repository includes the source code for the package, a Dockerfile to build the container image, and a WDL file that was used to run TotalReCall using Terra/Cromwell. Pre-built Docker images are published on Dockerhub and publicly available. A repository including Python notebooks to generate all analysis figures in this manuscript will also be made available on GitHub.

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

## Acknowledgements

The results shown here are primarily based upon data generated by the TCGA Research Network: https://www.cancer.gov/tcga. We would like to thank Nicole Rusk for editing the manuscript. This research was funded in part through the NIH/NCI Cancer Center Support Grant P30CA008748 (A.S., D.H., B.G.); NIH grants R01AI081848 (B.G.), R01CA240924 (A.S., D.H., B.G.), and U01CA228963 (A.S., D.H., B.G.); and the Mark Foundation ASPIRE award (B.G.). The authors would like to thank Kathy Burns, John LaCava, Agnel Sfeir, Martin Taylor, and David Ting for many helpful discussions. This work utilized resources from the High Performance Computing Group at Memorial Sloan Kettering Cancer Center. This work was supported by the Halvorsen Center for Computational Oncology at Memorial Sloan Kettering Cancer Center.

## Author contributions

B.D.G. and M.F. conceived and designed the study and provided resources to support the study. A.S. developed the TotalRecall algorithm and wrote the program and J.B. conducted the initial analyses, including statistical analyses and figure generation, and coded the analytical notebooks. B.D.G., M.F., J.B., L.D., A.S., A.W.D., E.G.R., E.B., and D.M.Z. contributed to the discussion of the results and methods. L.D., D.H., B.T., C.C., J.Z.Z., W.M., and C.A. contributed to data analyses and maintenance of the code base. J.K. and D.H. contributed to the discussion of LFS and p53 related results. J.B., A.S., M.F., B.D.G., and L.D. drafted the manuscript. L.D., B.D.G., M.F., and A.S. revised the manuscript. L.D., M.F. and B.D.G. co-supervised the work.

## Competing interests

At the time of the work, J.M.B., B.T., A.W.D., J.Z.Z., E.G.R., W.M., C.C., C.A., L.D., D.M.Z., and M.F. were full-time employees of, and hold stock options of, ROME Therapeutics. B.D.G. is a scientific co-founder of, consults for, and holds stock options of ROME Therapeutics. B.D.G. received honoraria from Merck, Bristol Meyers Squibb (BMS), and Chugai Pharmaceuticals; research funding from BMS and Merck; and has been a compensated consultant for Darwin Health, Merck, PMV Pharma, Shennon Biotechnologies, and Synteny.AI. A.S. has done consulting work for PMV Pharma and ROME Therapeutics; he holds stock options of ROME Therapeutics. E.B. is a full-time employee of The Broad Institute. J.K. spouse has received consulting fees from ROME Therapeutics, PanTher Therapeutics, Tekla Capital, abrdn, and Sonata Therapeutics; is a founder and has equity in ROME Therapeutics, PanTher Therapeutics and TellBio, Inc.; is on the advisory board for ImproveBio, Inc.; has received honorariums from AstraZeneca, Moderna and Ikena Oncology; and receives research support from ACD-Biotechne, AVA LifeScience GmbH, Incyte Pharmaceuticals, and Sanofi. The remaining authors declare no competing interests.
