## [Transparent Peer Review file · Nature Communications]

Pan-cancer multi-omic model of LINE-1 activity reveals locus heterogeneity of retrotransposition efficiency

Corresponding Author: Dr Benjamin Greenbaum

Version 0:

Reviewer comments:

Reviewer #2

(Remarks to the Author)

Behr et al. present an impressive amount of pan-cancer L1 analyses, greatly improving the manuscript with novel and exciting findings. The analysis of a newly expanded TCGA dataset brings statistical power to thoroughly confirm previous findings and provide insightful new signals. I'd like to praise their efforts to benchmark and improve L1 retrotransposition (RT) detection using their own method, TotalRecall, and xTea, and their rigor in testing the robustness of their findings regarding L1 RT. My initial comments and those of Reviewer 1, in my opinion, were fully addressed.

On the other hand, they present a large amount of new analytical data, including locus-specific L1 expression levels and RT efficiency. I agree with the overall variability of locus-specific RT expression and activity; however, they did not perform any benchmarking to confirm the accuracy of locus-specific L1 expression, simply relying on a tool published by another group. Additionally, the authors need to tone down their claims about RT efficiency, as the analysis is limited to insertions with transduction signals from reference L1s, which is only 2.5% of all insertions. Here are my comments to help improve the manuscript, including the two main concerns.

1. In the abstract, the authors summarize the dual role of P53 in L1 suppression as "both by disinhibiting L1 expression and not selecting against its reintegration." I don't understand what "not selecting against its reintegration" means and on what grounds they claim it.
2. The authors analyzed TCGA cancer samples with pathogenic germline p53 mutations, i.e., 21 LFS cases from 15 cancer types and found not higher, but similar levels of L1 expression and RT to non-LFS cancer samples. The sample size seems to be too small to draw a confident conclusion. If their finding is true, how is it compatible with higher L1 activity in cancers with somatic P53 mutations? It would be impossible given the already small sample size, but it would be more helpful to examine L1 expression of (adjacent) normal tissues from LFS cases. Furthermore, in both the abstract and the discussion, the authors mention the potential benefit of L1 therapeutics for LFS patients. Please elaborate on the rationale and background.
3. Some supplemental tables require column descriptions. For example, what do REF_REP_XT and DIST represent in table S3?
4. On page 3, the authors state that they have mapped 4870 TRTs ("transduction-bearing RTs") to genomic source coordinates and identified 726 unique source sites. They also mention that "the vast majority are thought to be polymorphic non-reference L1s". Could the authors provide the exact or approximate fractions of non-reference source L1s? And have the 726 unique sources been provided as a supplementary table, presumably in Table S3 under "TD_SRC"? What does "sibling" mean, for example, in "chr22:28670290~sibling" in the "TD_SRC" column of Table S3?
5. The accuracy of locus-specific L1 expression needs to be more convincingly demonstrated as it forms the basis of multiple downstream analyses and the title of the revised manuscript. RT-competent L1s are young with low sequence variability, which could lead to unreliable short-read RNA-seq read mapping. In contrast to their thorough benchmarking of

L1 RT detection, the authors simply relied on a published method without any further benchmarking, such as comparison with long-read RNA-seq. One indirect way of benchmarking could be to use expression measurements of the source L1s that generate TRTs. In fact, the authors compared "only" (L1 body) and "run-on" (L1 3' flank) expression of source TRT L1s and claimed a high correlation between them, referring to "Supplement for details" without mentioning specific figures or sections. I assume they were referring to Supplementary Fig. S3, which shows heatmaps of only two source L1s, Xp22.2-2 and 22q12.1. Why do the authors not examine all TRT source L1s and make a quantitative comparison? For each source, they could get "only" and "run-on" values, and check their correlation across all TRT sources?

6. The RT efficiency analysis presented is very limited to source L1s with identified TRTs from reference L1s. As the authors report, only ~15% of all L1 insertions are TRTs, and only 2.5% of all L1 insertions are TRTs from reference L1s, which were the subject of their RT efficiency analysis. Therefore, the authors must clearly state the limitations of their analysis. Given this limitation, along with the uncertain accuracy of locus-specific L1 expression, although the authors perform some additional analyses to justify their RT activity analyses, I think they need to call it "TRT efficiency" rather than "RT efficiency."

7. In Figure 5b, the authors state that "the four loci in the 9q21.32-2 cluster are more uniformly expressed across tumor types than any other group of loci." How is the uniformity defined? To the naked eye, there seem to be other loci with more uniform expression across cancer types. The authors need to provide quantitative evidence to support this statement.

8. Figure 3 legend states "121 loci with evidence of in vitro transductions." Shouldn't it be transcription or retrotransposition instead of "transduction"?

9. Why not mark the significance of differential expression of 121 known RT-competent loci between cancer and normal in Fig. 3b as they did for 1483 loci in Figure 3a?

10. Does the inversion rate differ across cancer types or remain consistent? This information could provide insights into the mechanisms underlying twin priming.

12. The clustering of 1483 source L1s subject to L1 efficient analysis is not clear. They described that source loci were assigned to the same cluster out of 13 final clusters if they shared assignments by both clusterings. One cluster named "1448 loci: 3p21.1-1, ..." contains 1,448 L1s out of the 1,283 loci (97.6%, why not out of 1483?). I doubt that the summary statistic from this gigantic cluster could represent any meaningful biology. Nevertheless, it would be helpful to provide more information about the two different clusterings, such as the clustering heatmap with the dendrogram and more explanation of the cluster assignment procedure.

13. In Figure 5d, the authors state that the bottom 6 locus clusters including the giant cluster "1448 loci: 3p21.1-1, ..." have discrepancies between L1 RNA and RT. But to the naked eye, the gigantic cluster "1448 loci: 3p21.1-1, ..." seems to show consistently low levels of expression and RT across cancer types.

14. On page 6 "Quantifying efficiency differences between L1 loci", they performed the linear regression analysis using total RTs per sample as the response variable to locus-level RNA. It is difficult to understand the rationale behind this analysis because the total RT is influenced by the activity of all active loci in the sample, not just by an individual TRT source tested. More explanation and rationale are needed. It is not surprising that the results do not provide any meaningful insights.

15. They analyzed 670 tumor-adjacent normal samples with RNA-seq and found no association between L1 expression and p53 mutations. They may want to describe the limitation of their analysis if L1 expression dysregulation might start in a focal area contributing to p53 mutation in cancer, and the normal tissue site sampled for RNA-seq might not retain the signal.

16. In the Discussion: they mentioned "while maintaining the sensitivity gains of the clipped read-based callers". This is misleading as other methods such as Tea and Traffic-mem also utilize clipped reads. Perhaps they could say gains of the methods that prioritize clipped reads over discordant reads, as increasing read lengths are likely to provide more signal than discordant reads.

17. In the Discussion, the authors mentioned that L1 RT burden is associated with worse clinical prognosis. But I cannot find relevant data to support this.

(Remarks on code availability)

The tool developed by the authors, TotalRecall is not yet available on GitHub, so it couldn't be tested. The authors mention it will be available upon publication along with other scripts.

Reviewer #3

(Remarks to the Author)

In this manuscript, Greenbaum and colleagues focused on the relationship between somatic L1 RNA expression and L1 insertions across 32 tumor types. Its locus specific analyses reveal substantial variation among intact L1 loci within and across tumor types. Compared to the initial submission, the revision has been significantly improved both in its analysis and in its interpretation.

Specific comments:

The Introduction section is atypical in its content and organization. The first paragraph is a brief background about L1. The second already talks about the new approach. The third again goes back to background about how L1 activity has been evaluated in the literature. The last paragraph covers some background about p53 and also their new approach/results. Unless a reader is very versed in literature about L1 omics in cancer, it can be challenging to evaluate the context in which this new report contributes to our understanding of L1 biology in cancer.

Results Section. At the end of first paragraph, the actual number of full-length L1 sequence should be indicated in parentheses.

The active genomic loci that are described at the bottom of the first page of Results Section should be provided in a supplemental table. Later analysis focused on 121 elements (second page), which should also be listed or indicated. Supplemental Table 6 has a list of loci. It is unclear how those are related to the number of loci described in the first two Results sections.

The third Results section describes “locus efficiency”. Here, a few related terms are used: locus efficiency, RT efficiency, and fitted coefficients. In the Discussion, “retrotransposition efficiency” is also used. There are also two models in the Methods section: Efficiency model and Total RT ~ locus RNA model. To an uneducated reader, these terms and models can be confusing. For example, why efficiency is used in the efficiency model but coefficient is used in the second model? Also how should the value be interpreted?

The fourth Results section is about the role of p53. The authors emphasize a model in which p53 independently regulates L1 RNA and L1 insertion. Can anything that cannot be explained by TP53 mutation be simply mediated by something else other than p53? In fact, the authors have found mutations in other genes in the subsequent section. It should also be emphasized that L1 RNA and L1 insertion are measurements at different time scale. They may not always correlate. So is L1 RNA and TP53 mutation.

The last Results section extends analysis to TP53 germline mutations, showing similar L1 activities in tumors patients with and without LFS. The significance of this finding seems overstated, with wording like “for the first time” both in the abstract and introduction.

In the Results section, there are multiple references to “Supplement” without specifying which file, table or figure, making it hard to follow.

At the end of the Discussion section, the authors state “L1 RT burden seemed to be significantly associated with a worse clinical prognosis”. Where are the corresponding data?

(Remarks on code availability)

Version 1:

Reviewer comments:

Reviewer #2

(Remarks to the Author)

All my previous comments have been successfully addressed. The only remaining, but crucial, suggestion is for the authors to deposit their germline transposon insertion calls. Given the significant effort and resources that went into generating this call set using two different callers, it would be a highly valuable resource. Making this dataset available through controlled access on the GDC data portal would contribute significantly to the advancement of transposon research.

(Remarks on code availability)

Reviewer #3

(Remarks to the Author)

The authors have adequately addressed my comments.

(Remarks on code availability)

We thank you for the consideration of our work. We would also like to thank the reviewers for their thorough read-throughs and comments, which we believe have helped to improve upon the originally submitted manuscript.

We have addressed each of the comments raised in a point-by-point response, attached, and added new tables and figures as necessary. In particular, we have updated the following:

- Significantly clarified language around, and description of, the various classifications of LINE-1 loci referred to in our analyses (e.g., defining which are “active”, and in which context). Along these lines, we have also worked to better delineate where our analyses specifically refer to “transduction-bearing RTs” compared to “RTs” more broadly.
- We have included a new set of simulations which compares the performance of L1EM, which is the software primarily used in this manuscript, and TElocal, which is another publicly available software, in performing LINE-1 locus-level expression quantification. These simulations are included in the Extended Methods and referenced in the manuscript.
- We have added supplementary tables to include relevant reference data to aid in the reproducibility of our analyses, including the specific list of 121 “active” loci which we refer to multiple times throughout the manuscript.
- Finally, we have newly included an Extended Methods document, which was unintentionally omitted from our original submission. The Extended Methods document provides significantly more detail on the methodology referred to throughout the manuscript, including that related to calling retrotranspositions, validation of RT detection with long reads, locus clustering, p53 mediation modeling, and clinical stratification analysis.

We believe that with the additions made above, the reviewers and editors will find the story and results presented in “Pan-cancer multi-omic model of LINE-1 activity reveals locus heterogeneity of retrotransposition efficiency” a significant and compelling contribution to the field. We have highlight additions to the main text in light of reviewer comments in yellow.

Reviewer #2

Behr et al. present an impressive amount of pan-cancer L1 analyses, greatly improving the manuscript with novel and exciting findings. The analysis of a newly expanded TCGA dataset brings statistical power to thoroughly confirm previous findings and provide insightful new signals. I'd like to praise their efforts to benchmark and improve L1 retrotransposition (RT) detection using their own method, TotalRecall, and xTea, and their rigor in testing the robustness of their findings regarding L1 RT. My initial comments and those of Reviewer 1, in my opinion, were fully addressed.

On the other hand, they present a large amount of new analytical data, including locus-specific L1 expression levels and RT efficiency. I agree with the overall variability of locus-specific RT expression and activity; however, they did not perform any benchmarking to confirm the accuracy of locus-specific L1 expression, simply relying on a tool published by another group. Additionally, the authors need to tone down their claims about RT efficiency, as the analysis is limited to insertions with transduction signals from reference L1s, which is only 2.5% of all insertions. Here are my comments to help improve the manuscript, including the two main concerns.

1. In the abstract, the authors summarize the dual role of P53 in L1 suppression as “both by disinhibiting L1 expression and not selecting against its reintegration.” I don't understand what “not selecting against its reintegration” means and on what grounds they claim it.

We have clarified the language in this sentence.

2. The authors analyzed TCGA cancer samples with pathogenic germline p53 mutations, i.e., 21 LFS cases from 15 cancer types and found not higher, but similar levels of L1 expression and RT to non-LFS cancer samples. The sample size seems to be too small to draw a confident conclusion. If their finding is true, how is it compatible with higher L1 activity in cancers with somatic P53 mutations? It would be impossible given the already small sample size, but it would be more helpful to examine L1 expression of (adjacent) normal tissues from LFS cases. Furthermore, in both the abstract and the discussion, the authors mention the potential benefit of L1 therapeutics for LFS patients. Please elaborate on the rationale and background.

We appreciate the reviewer's insightful comments. While we, and others, have demonstrated that TP53 is a key regulator of L1 activity, it is important to note that there are several other gatekeepers for genome stability, along with various layers

of repression that influence L1 retrotransposon activity, such as methylation status and loss of TP53 heterozygosity.

While we agree that it would be interesting to look at expression from adjacent normal tissue, unfortunately, no matched adjacent normal RNA-seq samples are available for the 21 LFS cases on the GDC data portal (<https://portal.gdc.cancer.gov>). All matched control samples were derived from blood, not adjacent normal tissues (through a manual check of portal.gdc.cancer.gov) which limits our ability to perform the suggested analysis.

We have also revised our statements regarding the potential therapeutic opportunities of using L1 inhibitors to treat LFS patients in both the abstract and discussion sections to better reflect these considerations.

3. Some supplemental tables require column descriptions. For example, what do REF_REP_XT and DIST represent in table S3?

In addition to the supplementary table captions, which describe the columns, we have also included a Data Dictionary tab mapping out descriptors for columns in the supplementary tables for those tables where columns are not clear.

4. On page 3, the authors state that they have mapped 4870 TRTs ("transduction-bearing RTs") to genomic source coordinates and identified 726 unique source sites. They also mention that "the vast majority are thought to be polymorphic non-reference L1s". Could the authors provide the exact or approximate fractions of non-reference source L1s? And have the 726 unique sources been provided as a supplementary table, presumably in Table S3 under "TD_SRC"? What does "sibling" mean, for example, in "chr22:28670290~sibling" in the "TD_SRC" column of Table S3?

We added the 726 source elements to Supplementary Table 4. For each source L1, we provided the number of offspring L1s generated from this source L1, as well as the annotation of source L1 is polymorphic or reference L1. Out of the 726 source L1s, 610 and 116 are polymorphic and reference copies, respectively. We have included these exact numbers in the manuscript as suggested.

"Sibling" TRTs are one type of transduction originally introduced in the xTea manuscript, defined as those TRTs whose transduction sequences can be uniquely mapped to a region where no reference or polymorphic full-length LINE-1 is annotated. In practice, we found some of the canonical transductions were wrongly

annotated to “sibling” subtype, thus we rescued some of those are close to known reference full-length L1s (Table S3; column “source_l1em_locus”). The rest of the “sibling” transductions were not counted when estimating the LINE-1 retrotransposition activity.

We have added some clarifying language in the Data Dictionary of Supplementary Table 3 to this effect.

5. The accuracy of locus-specific L1 expression needs to be more convincingly demonstrated as it forms the basis of multiple downstream analyses and the title of the revised manuscript. RT-competent L1s are young with low sequence variability, which could lead to unreliable short-read RNA-seq read mapping. In contrast to their thorough benchmarking of L1 RT detection, the authors simply relied on a published method without any further benchmarking, such as comparison with long-read RNA-seq. One indirect way of benchmarking could be to use expression measurements of the source L1s that generate TRTs. In fact, the authors compared "only" (L1 body) and "run-on" (L1 3' flank) expression of source TRT L1s and claimed a high correlation between them, referring to “Supplement for details” without mentioning specific figures or sections. I assume they were referring to Supplementary Fig. S3, which shows heatmaps of only two source L1s, Xp22.2-2 and 22q12.1. Why do the authors not examine all TRT source L1s and make a quantitative comparison? For each source, they could get "only" and "run-on" values, and check their correlation across all TRT sources?

We initially elected to use L1EM as it has been previously shown to have good locus-level quantification accuracy (see McKerrow et al.). For a more robust estimation of the accuracy of L1EM at LINE-1 locus-level quantification, we now also present a set of simulations and compare the accuracy of locus-level calls between L1EM and another commonly used tool, TELocal, demonstrating the performance of L1EM. We have included a description of these simulations and their results in the Extended Methods and added a reference to these simulations in the main text.

Additionally, we appreciate the reviewer for pointing out missing reference information and have updated references to “Supplement” throughout the manuscript, as well as included an Extended Methods document that was missing from the previous submission. In this instance, the correct reference was to Supplementary Figure 6, where we show boxplots of L1 expression across TCGA for

the full set of LINE-1 loci as well as the subset of 121 active loci, for “run-on” transcripts, showing that these produce similar results as stated in the main text. Given the results of the updated simulations provided, we feel this is sufficient to address the question of locus-level quantification without additional comparisons of “only” and “run-on” values at the locus level.

6. The RT efficiency analysis presented is very limited to source L1s with identified TRTs from reference L1s. As the authors report, only ~15% of all L1 insertions are TRTs, and only 2.5% of all L1 insertions are TRTs from reference L1s, which were the subject of their RT efficiency analysis. Therefore, the authors must clearly state the limitations of their analysis. Given this limitation, along with the uncertain accuracy of locus-specific L1 expression, although the authors perform some additional analyses to justify their RT activity analyses, I think they need to call it "TRT efficiency" rather than "RT efficiency."

We agree with the reviewer that the locus-level efficiency analyses are, by necessity, analyses of TRT rather than RT events, given that RT events at the locus level can only be captured for LINE-1 loci that generate transductions.

In paragraph 2 of the section, “Locus-level analysis of L1”, we specifically refer to the “landscape of locus-TRT burden” with respect to these analyses and attempt to make clear where we are referring specifically to the TRT burden and when we refer to RT events in general.

Further, given the high correlation between our predicted efficiencies and those obtained experimentally *in vitro* in Brouha et al., as referenced in the Discussion, it seems that TRT burden may be a reasonable approximation of RT burden for loci which generate TRTs.

However, to make this clearer, and particularly the part about loci which commonly generate TRTs, we have added text in the discussion to point out the limitation of this analysis (see specifically paragraph 5 in the Discussion).

7. In Figure 5b, the authors state that "the four loci in the 9q21.32-2 cluster are more uniformly expressed across tumor types than any other group of loci." How is the uniformity defined? To the naked eye, there seem to be other loci with more uniform expression across cancer types. The authors need to provide quantitative evidence to support this statement.

We thank the reviewer for identifying this discrepancy. We have updated this analysis and defined variability as follows: We measured cluster variability by counting the number of subtypes in which the cluster is expressed as defined by the mean expression being ≥ 0.1 TPM. We grouped clusters into 3 bins labeled “high”, “medium” and “low”, depending on if the clusters are expressed in >15 subtypes, between 5 and 15 or below 5, respectively. We have updated the figure to reflect this and updated the figure caption to describe how the bins were calculated.

With this updated definition, our previous bins of variable, uniform, and minimal expression still hold with the exception of one cluster. However, the statement the reviewer alludes to is indeed incorrect, and we have removed this from the text.

8. Figure 3 legend states “121 loci with evidence of in vitro transductions.” Shouldn’t it be transcription or retrotransposition instead of “transduction”?

We thank the reviewer for pointing out this discrepancy and general unclear language around this set of 121 loci. The referenced Ebert paper’s Table S23 includes loci which are shown to retrotranspose *in vitro*. Our set of 121 are a combination of those loci and those with evidence of transductions in our analyses, as referenced in the third paragraph of “High L1 RNA corresponds to high L1 RT burden”:

Expression analysis of the subset of 121 elements for which there is evidence of retrotransposition from either our analysis (87 elements) or previously published studies (74 total, including 34 additional²⁷) yielded similar results (Fig. 3b).

We have updated the language in manuscript to be clearer:

“We also make use of a subset of 121 L1HS and L1PA2 elements in Figure 3 and Extended Data Figures 2 and 5, which is based on the combination of all source elements of transductions identified in this study and elements with in vitro evidence of retrotransposition as previously annotated in Supplemental Table 23 of Ebert et al”

We also update the caption in Figure 3 to reflect the above language.

We further clarified language throughout the manuscript to this end, referring to this set of 121 loci as “RT-competent”

9. Why not mark the significance of differential expression of 121 known RT-competent loci between cancer and normal in Fig. 3b as they did for 1483 loci in Figure 3a?

We thank the reviewer for noticing this omission. We have added the significance values in Figure 3b.

10. Does the inversion rate differ across cancer types or remain consistent? This information could provide insights into the mechanisms underlying twin priming.

This is an interesting question. We investigated this as suggested and found that the inversion rate is relatively constant across tumor types. We have added this result to the manuscript (Figure 2b).

11. The clustering of 1483 source L1s subject to L1 efficient analysis is not clear. They described that source loci were assigned to the same cluster out of 13 final clusters if they shared assignments by both clusterings. One cluster named "1448 loci: 3p21.1-1, ..." contains 1,448 L1s out of the 1,283 loci (97.6%, why not out of 1483?). I doubt that the summary statistic from this gigantic cluster could represent any meaningful biology. Nevertheless, it would be helpful to provide more information about the two different clusterings, such as the clustering heatmap with the dendrogram and more explanation of the cluster assignment procedure.

While we agree that the large cluster of 1448 loci is itself unlikely to provide any insight into the mechanisms of LINE-1 activity, we took this clustering approach as described in the Methods because it is algorithmically unbiased. The result is a set of loci which captures those with low expression and low RT activity. This seems reasonable given that this analysis includes all L1HS and L1PA2 loci, of which we know only a small fraction are full-length and intact, and a similarly small fraction have been observed *in vitro* to produce TRTs (see the repeated analysis for the latter in Extended Data Fig. 5).

We have also expanded the explanation of the clustering methodology in the methods as suggested and included both the RNA-based clustering as well as the RT-based clustering results in the Extended Methods. Generally speaking, what the more fine-grained analysis of the clustering indicates is that loci that fall outside of the large low-RNA **and** low-TRT cluster still typically fall into the category of **either** low RNA and higher TRT **or** low TRT and higher RNA, with only one cluster composed

of loci which neither fall into the lowest TRT nor lowest RNA individual clusters. These results have been added to the Extended Methods.

12. In Figure 5d, the authors state that the bottom 6 locus is including the giant cluster "1448 loci: 3p21.1-1, ..." have discrepancies between L1 RNA and RT. But to the naked eye, the gigantic cluster "1448 loci: 3p21.1-1, ..." seems to show consistently low levels of expression and RT across cancer types.

We agree with the reviewer and updated language in the manuscript (paragraph 4 in section "Locus-level analysis of L1").

13. On page 6 "Quantifying efficiency differences between L1 loci", they performed the linear regression analysis using total RTs per sample as the response variable to locus-level RNA. It is difficult to understand the rationale behind this analysis because the total RT is influenced by the activity of all active loci in the sample, not just by an individual TRT source tested. More explanation and rationale are needed. It is not surprising that the results do not provide any meaningful insights.

Our initial analysis indeed was focused on TRT events regressed against locus-level expression, for the 48 loci for which TRTs were detected in at least 2 tumors. However, as we stated in the main text, one of the limitations of such an analysis is that TRTs represent only a small fraction of the total RT call set. The reason we took the total RTs per sample as the response variable in this secondary analysis was to try to address this limitation.

We take the reviewer's suggestion that multiple loci may contribute to RT activity in different ways, with some providing RT substrate while others provide the RT machinery, and therefore that the regression analysis proposed cannot adequately resolve which loci are doing what. Here we point out that the *cis* preference for LINE-1 activity has been previously reported (see Kulpa and Moran, *Nat Struct Mol Biol.* 2006; added to references). So, while we agree that this is a limitation of the analysis and have added language to expressly point this out, we believe it is still a reasonable approach to take.

14. They analyzed 670 tumor-adjacent normal samples with RNA-seq and found no association between L1 expression and p53 mutations. They may want to describe the limitation of their analysis if L1 expression dysregulation might start in a focal area contributing to p53 mutation in cancer, and the normal tissue site sampled for RNA-seq might not retain the signal.

We thank the reviewer for the comment and have incorporated language speaking to the limitation of this analysis.

15. In the Discussion: they mentioned "while maintaining the sensitivity gains of the clipped read-based callers". This is misleading as other methods such as Tea and Traffic-mem also utilize clipped reads. Perhaps they could say gains of the methods that prioritize clipped reads over discordant reads, as increasing read lengths are likely to provide more signal than discordant reads.

We thank the reviewer for the suggestion and have incorporated the suggested language.

16. In the Discussion, the authors mentioned that L1 RT burden is associated with worse clinical prognosis. But I cannot find relevant data to support this.

We apologize for the omission and thank the reviewer for catching this. These results were included in Extended Methods and Supplemental Figure 11, which are now both included.

Reviewer #2 (Remarks on code availability):

The tool developed by the authors, TotalRecall is not yet available on GitHub, so it couldn't be tested. The authors mention it will be available upon publication along with other scripts.

We have made the code for TotalRecall available (<https://github.com/Rome-Tx/totalrecall>). Included also is a docker image for ease of use.

Reviewer #3

In this manuscript, Greenbaum and colleagues focused on the relationship between somatic L1 RNA expression and L1 insertions across 32 tumor types. Its locus specific analyses reveal substantial variation among intact L1 loci within and across tumor types. Compared to the initial submission, the revision has been significantly improved both in its analysis and in its interpretation.

Specific comments:

1. The Introduction section is atypical in its content and organization. The first paragraph is a brief background about L1. The second already talks about the new approach. The third again goes back to background about how L1 activity has been evaluated in the literature. The last paragraph covers some background about p53 and also their new approach/results. Unless a reader is very versed in literature about L1 omics in cancer, it can be challenging to evaluate the context in which this new report contributes to our understanding of L1 biology in cancer.

We thank the reviewer for pointing this out and have restructured the introduction to improve the organization and flow.

2. Results Section. At the end of first paragraph, the actual number of full-length L1 sequence should be indicated in parentheses.

We have updated the text and included the actual number of full-length L1 sequences as suggested (N=2143).

3. The active genomic loci that are described at the bottom of the first page of Results Section should be provided in a supplemental table. Later analysis focused on 121 elements (second page), which should also be listed or indicated. Supplemental Table 6 has a list of loci. It is unclear how those are related to the number of loci described in the first two Results sections.

We have included these loci as a supplementary table, as suggested. Supplemental table 6 includes loci for which we observe transductions in our analyses, and additionally includes efficiency models for those loci, and therefore are not entirely the same as the set of 121 “active” loci mentioned.

4. The third Results section describes “locus efficiency”. Here, a few related terms are used: locus efficiency, RT efficiency, and fitted coefficients. In the Discussion, “retrotransposition efficiency” is also used. There are also two models in the Methods section: Efficiency model and Total RT ~ locus RNA model. To an uneducated reader, these terms and models can be confusing. For example, why

efficiency is used in the efficiency model but coefficient is used in the second model? Also how should the value be interpreted?

We thank the reviewer for this comment and, upon revisiting the writing in the section, agree that the mixed use of terminology is confusing. All of these terms essentially refer to the same concept, which is the rate at which TRT events are observed given the corresponding level of RNA observed, but we use “efficiencies” specifically for the per-locus analysis and “fitted coefficients” in other contexts due to the more general nature of the other analyses (such as that with total RT as a response variable). We have provided more explanation and interpretation of these terms, as well as more context on the interpretation of the efficiency value, and hope that these changes improve the clarity of the results.

5. The fourth Results section is about the role of p53. The authors emphasize a model in which p53 independently regulates L1 RNA and L1 insertion. Can anything that cannot be explained by TP53 mutation be simply mediated by something else other than p53? In fact, the authors have found mutations in other genes in the subsequent section. It should also be emphasized that L1 RNA and L1 insertion are measurements at different time scale. They may not always correlate. So is L1 RNA and TP53 mutation.

To the reviewer’s point, there are multiple known mechanisms involved in the regulation of L1 RNA and retrotransposition which are p53 independent. The fact that the correlation between p53 mutation status and LINE-1 RT is ~ 0.4 (Figure 7b) supports p53-independent regulatory mechanisms. The purpose of the mediation model was to assess the specific hypothesis that LINE-1 RT regulation by p53 can be wholly explained by p53 regulation of LINE-1 RNA expression, given the degree of correlation we observed between RNA and RT activity, which has not previously been investigated.

We have also added language pointing out the differences in time scale between L1 RNA and RT, as well as p53 mutation.

6. The last Results section extends analysis to TP53 germline mutations, showing similar L1 activities in tumors patients with and without LFS. The significance of this finding seems overstated, with wording like “for the first time” both in the abstract and introduction.

We have removed the wording “for the first time” from the section and mention only that the increased sample size allowed the interrogation of this rare phenotype.

7. In the Results section, there are multiple references to “Supplement” without specifying which file, table or figure, making it hard to follow.

We thank the reviewer for this observation, which pointed us to an Extended Methods document that was unintentionally omitted from the original submission. We have updated and included this document and clarified all references to “Supplement” throughout the text, so that we only refer to specific sections of the Extended Data and Supplemental Figures.

8. At the end of the Discussion section, the authors state “L1 RT burden seemed to be significantly associated with a worse clinical prognosis”. Where are the corresponding data?

We apologize for the omission and thank the reviewer for catching this. These results were included in Extended Methods and Supplemental Figure 11, which are now both included.

We thank both Reviewers for their comments and appreciate the opportunity to publish our work in Nature Communications, which has greatly benefitted from their feedback.

Reviewer #2 (Remarks to the Author):

All my previous comments have been successfully addressed. The only remaining, but crucial, suggestion is for the authors to deposit their germline transposon insertion calls. Given the significant effort and resources that went into generating this call set using two different callers, it would be a highly valuable resource. Making this dataset available through controlled access on the GDC data portal would contribute significantly to the advancement of transposon research.

We thank the reviewer for their comments. Our call set did not call the germline directly, such as via an exhaustive comparison of germline versus the reference genome which would take a great deal of time. We did however annotate all genomic calls which have germline support. We now plot the distribution of such “pseudo” germline calls in Extended Data Figure 1 a-c (pasted below), along with their length distribution and allele frequencies (where they follow an expectedly neutral distribution). We are in the process of working with dbGAP to upload which will take a few weeks.

Reviewer #3 (Remarks to the Author):

The authors have adequately addressed my comments.

We sincerely appreciate the reviewer's feedback.